# Demonstration of a beam loaded nanocoulomb-class laser wakefield accelerator

J.P. Couperus [1,2], R. Pausch [1,2], A. Köhler [1,2], O. Zarini[1,2], J.M. Krämer[1,2], M. Garten [1,2], A. Huebl [1,2], R. Gebhardt[1], U. Helbig[1], S. Bock[1], K. Zeil[1], A. Debus [1], M. Bussmann[1], U. Schramm [1,2] & A. Irman[1]

Laser-plasma wakefield accelerators have seen tremendous progress, now capable of producing quasi-monoenergetic electron beams in the GeV energy range with few-femtoseconds bunch duration. Scaling these accelerators to the nanocoulomb range would yield hundreds of kiloamperes peak current and stimulate the next generation of radiation sources covering high-field THz, high-brightness X-ray and $\gamma$-ray sources, compact free-electron lasers and laboratory-size beam-driven plasma accelerators. However, accelerators generating such currents operate in the beam loading regime where the accelerating field is strongly modified by the self-fields of the injected bunch, potentially deteriorating key beam parameters. Here we demonstrate that, if appropriately controlled, the beam loading effect can be employed to improve the accelerator's performance. Self-truncated ionization injection enables loading of unprecedented charges of $\sim$0.5 nC within a mono-energetic peak. As the energy balance is reached, we show that the accelerator operates at the theoretically predicted optimal loading condition and the final energy spread is minimized.

[1] Helmholtz-Zentrum Dresden - Rossendorf, Institute of Radiation Physics, Bautzner Landstrasse 400, 01328 Dresden, Germany. [2] Technische Universität Dresden, 01062 Dresden, Germany. Correspondence and requests for materials should be addressed to J.P.C. (email: j.couperus@hzdr.de) or to A.I. (email: a.irman@hzdr.de)

Electron-driven light sources have become indispensable tools for a broad range of fundamental and applied research. Large-scale facilities such as synchrotrons and free-electron lasers (FELs) provide extremely bright and short wavelength radiation, enabling microscopy at atomic resolution with femtosecond to attosecond time scales. However, conventional radio-frequency electron accelerators currently used to drive such sources are limited to only a few-kiloampere peak current. In contrast, the ultra-high accelerating field gradient in laser-plasma wakefield accelerators can sustain higher bunch charges within a few-femtosecond pulses enabling peak currents up to two orders of magnitude larger than found in conventional accelerators. Knowledge to operate laser-plasma accelerators in the high-current regime would make them ideal drivers for next-generation compact light sources covering high-field THz[1, 2], high-brightness X-ray[3, 4] and γ-ray[5, 6] sources, compact FELs[7–11] and laboratory-size beam-driven plasma accelerators[12, 13].

In laser-plasma wakefield acceleration, an ultrashort laser pulse propagating through an optically transparent plasma excites a plasma wake by the laser ponderomotive force[14]. For a sufficiently high driver laser intensity, plasma electrons are expelled from the laser vicinity, thereby creating a co-propagating near-spherical ion cavity[15]. In this so-called bubble or blow-out regime the accelerating wakefield can exceed several hundreds of GV m$^{-1}$. In contrast to conventional accelerators, electrons from the plasma background can be self-injected into the wakefield, and therefore no external electron source is required. Electrons that are injected into the right phase of this wakefield will be accelerated to high energies within a quasi-monoenergetic peak. Even though electron beams reaching into the multi-GeV energy range and few-femtosecond bunch durations have already been demonstrated[16–19], bunch charge has been limited to only a few tens of picocoulomb. Scaling the charge to the nanocoulomb range following original predictions[15] would yield hundreds of kiloamperes peak-current beams, the key component for next-generation compact light sources. During the submission of this manuscript, Li et al.[20] demonstrated generation of electron beams up to 20 kA from laser wakefield acceleration yet of a continuous spectrum extending up to 0.6 GeV. Laser-plasma accelerators generating such high currents accumulate enough charge such that the self-fields of the bunch will superimpose on the wakefield[21–23]. As a consequence the plasma cavity structure will be reshaped and the effective accelerating field along the bunch length will be modified affecting the final beam parameters, i.e., maximum energy and energy spread. This phenomenon is generally known as beam loading.

Lu et al.[22] estimated the number of particles that can be loaded into a three-dimensional (3D) nonlinear wake to scale with the normalized volume of the plasma bubble or the square root of the laser power. The same scaling was also found by Gordienko and Pukhov[21] with a different coefficient derived from simulations. The linear theory for beam loading in plasma accelerators was developed by Katsouleas et al.[24] In this one-dimensional approach it was found that an appropriate bunch shaping can minimize the energy spread typically associated with plasma accelerators. In recent work by Tzoufras et al.[23, 25] this theory was expanded for the 3D nonlinear case and an optimum trapezoidal bunch shape was investigated in order to efficiently convert the energy available in the wake into kinetic energy of electrons. In this ideal case, the accelerating field of the wakefield $E_z$ becomes constant ($E_s$) along the bunch and scales as:

$$\frac{Q_s}{1\text{nC}}\frac{eE_s}{m_e c \omega_p} \simeq 0.047\sqrt{\frac{10^{16}\text{cm}^{-3}}{n_p}}\left(k_p R_b\right)^4 \qquad (1)$$

where $c$ is the speed of light, $Q_s$ is the total loaded charge, $m_e$ and $e$ are the mass and charge of an electron and $n_p$ and $\omega_p \sim \sqrt{n_p}$ are the density and frequency of the plasma with the plasma wavenumber $k_p$. For a matched laser condition[22], $k_p R_b$ amounts to approximately $2\sqrt{a_0}$ with the bubble radius $R_b$ and the laser normalized vector potential $a_0$. Eq. (1) represents the optimal loaded case where there is a balance between the amount of loaded charge and the accelerating field. Here, all electrons in the bunch experience an identical accelerating field strength such that no energy spread is gained during the acceleration process. This leads to a bunch with minimum energy spread. From Eq. (1) the optimal loaded charge depends on the laser peak power $P$ as $Q_s \propto \sqrt{P}$ but is independent from the plasma density. However, the accelerating field $E_s$ and thus the achievable electron energy follows a $n_p^{2/3}$ dependency (for details see Supplementary Note 1).

Although indications of beam loading have been reported earlier[26, 27], no experimental studies exist for the case of a quasi-monoenergetic bunch in a heavily loaded wakefield. The work presented here is the first investigation that systematically explores the beam loading effect in the bubble regime and its consequences to the final beam quality over a large and well-controlled parameter range. We inject several hundred pC of charge into the bubble cavity while retaining a narrow energy spread. For this purpose a tailored scheme of the self-truncated ionization injection (STII)[28, 29] process is used (see Methods). This scheme relies on ionization injection, where the helium gas based acceleration medium is doped with a small fraction of

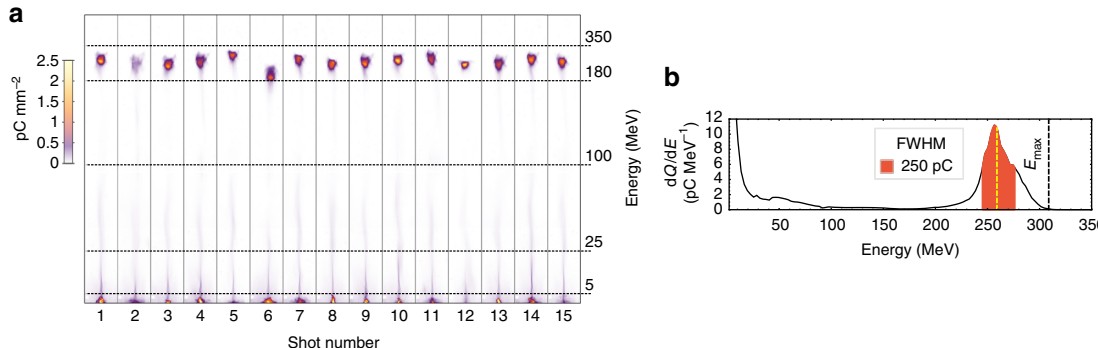

**Fig. 1** Energy spectra of 15 consecutive shots. **a** Raw energy electron spectra. The color map represents the charge density (pC mm$^{-2}$) on the detector. **b** Energy spectrum of the first shot from **a**. The *filled area* represents the charge within the FWHM, the *yellow dashed line* represents the mean peak energy and the *black dashed line* represents the maximum attained energy ($E_{max}$) at 0.1 pC MeV$^{-1}$. Obtained with a supersonic gas jet with a 1.6 mm-long plasma density plateau of $3.1 \times 10^{18}$ cm$^{-3}$, 1% nitrogen doping and 2.5 J laser energy in 30 fs FWHM duration. Line graphs of all shots shown in (**a**) can be found in Supplementary Fig. 2

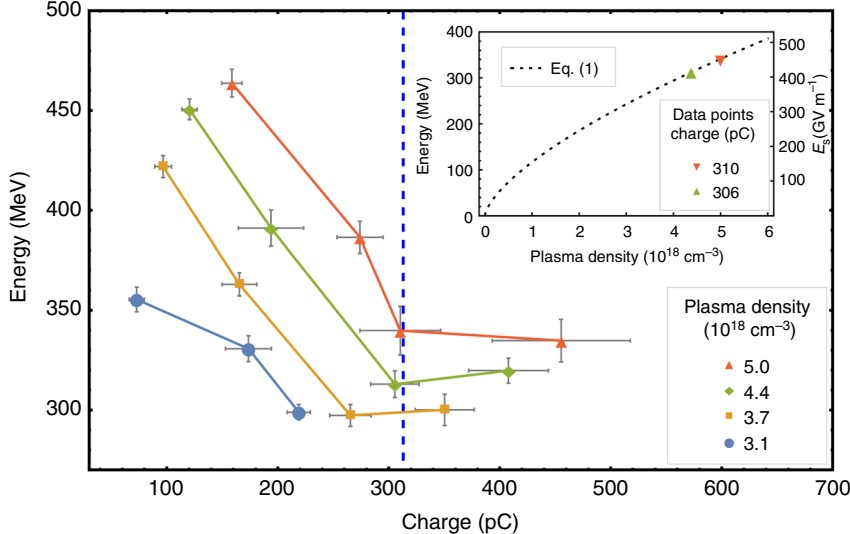

**Fig. 2** Electron energy dependency on both charge and plasma density. Connected data points show a set of equal plasma density, increasing charge with increasing nitrogen doping (see Supplementary Fig. 5). Electron energy is the $E_{max}$ and charge within the FWHM of the energy peak is displayed. The error bars represent the s.e.m. The estimated optimal loaded charge according to Eq. (1) of 313 pC is indicated by the *dashed vertical line*. The *inset* shows the predicted acceleration gradient (*right axis*) according to Eq. (1) (dotted line) for a laser peak power $P = 64$ TW (equivalent to the experiment) under optimal loading conditions. The *left axis* shows the predicted electron energy, taking 0.8 mm effective acceleration distance. Data points represent measured data points at the predicted optimum

nitrogen gas whose K-shell electrons will be ionized and subsequently trapped only near the intensity peak of the laser pulse[30, 31]. In the STII scheme laser and wakefield evolution restrict the time of electron injection into the wakefield, thus limiting the energy spread. The number of injected electrons can be tuned by varying the nitrogen doping concentration without changing the driver laser and plasma density parameters. Thus, in contrary to the work in ref. [26], the initial position and radius for trapping can be kept equal for various loaded charges. Loading-dependent injection length contributes only weakly to the energy spread (see Supplementary Note 4). Therefore, it enables us to minimize the interplay between the beam loading effect and the initial injection volume to the evolution of energy spread.

## Results

**Accelerator performance**. The typical performance of the accelerator is presented in Fig. 1a, showing raw data from the electron spectrometer for 15 consecutive shots with Fig. 1b showing a *line graph* of the first shot with a graphic representation of relevant beam parameters. This specific set was obtained for a 1.6 mm-long plasma density plateau of $3.1 \times 10^{18}$ cm$^{-3}$, 1% nitrogen doping and 2.5 J laser energy in 30 fs full-width at half-maximum (FWHM) duration. The mean peak energy of the electron beam is 250 MeV with 9% shot-to-shot s.d. The mean absolute energy spread (FWHM) is 36 MeV yielding a relative energy spread of 14%. The accelerator delivers an unprecedented charge within the peak (FWHM) of ~220 pC ± 40 pC shot-to-shot s.d. with a divergence of 7 mrad. The excellent shot-to-shot reproducibility gives the ability to perform statistical analysis over multiple shots for each data set belonging to a specific experimental parameter.

**Beam loading effects**. The influence of both injected charge and plasma density is clearly seen in the maximum attainable energy as shown in Fig. 2. Each data point represents the mean value from a set of up to 20 consecutive shots. For equal plasma density, indicated by connected data points, the injected charge was controlled by tuning the nitrogen doping between 0.2% and 1.5%

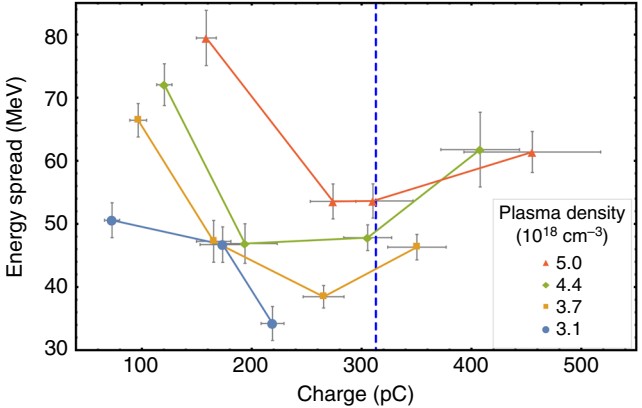

**Fig. 3** Beam absolute energy spread dependency on charge. Shown is the FWHM energy spread. The *dashed vertical line* represents the optimal load expected from Eq. (1). The *error bars* represent the s.e.m. A graph showing the relative energy spread can be found in Supplementary Fig. 7

(see Supplementary Fig. 5). With increasing injected charge, a decrease in maximum electron energy ($E_{max}$) is observed. Since the injection dynamics do not change significantly, this energy reduction indicates accelerating field suppression due to the load.

Beam loading perturbs the wakefield such that the accelerating field strength experienced by the trailing electrons is reduced. In our measurements we have observed a smaller decrease on the mean energy of electrons compared to the maximum attained energy discussed above (see Supplementary Fig. 6). Consequently, the beam energy spread narrows for higher loaded charges, as seen in Fig. 3. At the optimum load, flattening of the accelerating field is expected which then leads to a minimum of the energy spread. This effect is seen at a charge of ~300 pC in FWHM for all sets of plasma densities. This is in very good agreement with the value estimated by Eq. (1) marked by the *vertical line* in Figs. 2 and 3. Deviation from the optimum, either by loading less or more charge into the wakefield, leads to an increase in energy spread.

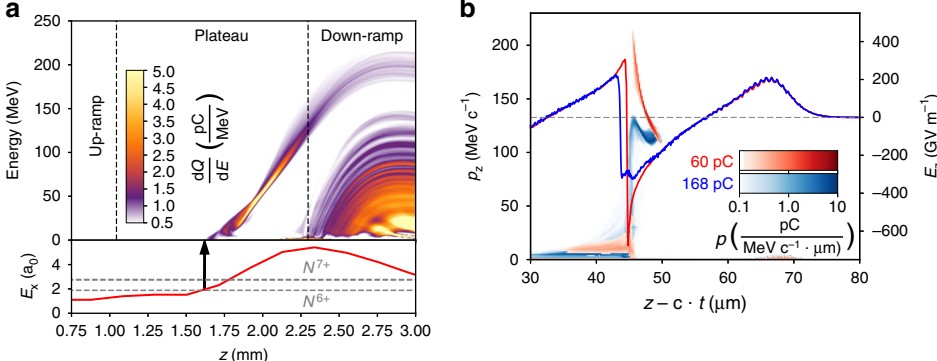

**Fig. 4** Energy evolution and beam loading effects during the acceleration process. Results from PIC simulations. **a** Energy histogram (*top*) showing evolution of the electron energy throughout the acceleration process. Final injected charge in the peak is 60 pC. The *bottom plot* shows the on-axis laser strength evolution with the *dotted lines* representing the required field for ionization of the two nitrogen K-shell electrons. Additional injection of both helium and nitrogen electrons occurs in the density down-ramp of the gas jet, resulting in a low-energy background in the final energy spectrum. **b** The effect of beam loading on the accelerating field $E_z$ (*line graphs*, *right axis*) and electron phase space (*color scale*, *left axis*) for 60 pC load in the peak (*red*) and 168 pC load (*blue*). Corresponding to $z = 2.3$ mm in **a**

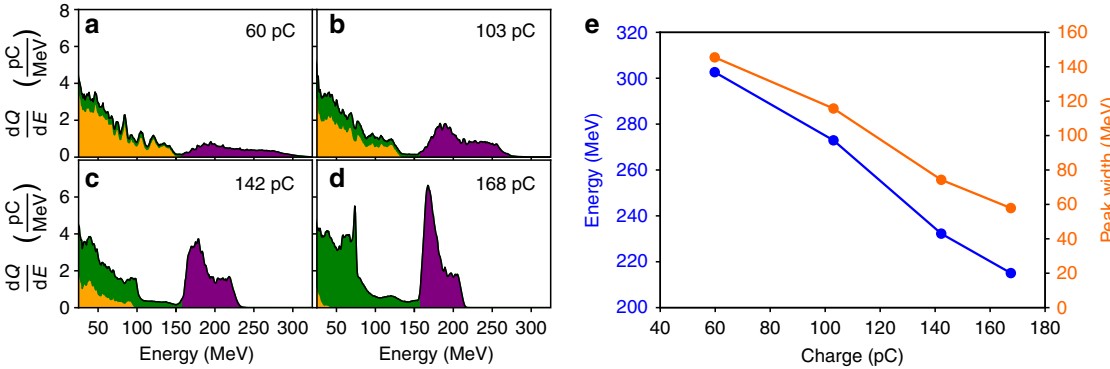

**Fig. 5** PIC simulation results illustrating beam loading effects. **a–d** Electron energy histograms for increasing injected charges within the energy peak (*purple area*). The *green-* and *purple-shaded area* is the contribution from electrons originating from nitrogen, and *orange-shaded area* from helium electrons. The *black line* indicates the cumulative spectrum. **a** Corresponds to the histogram seen in Fig. 4a. **e** Shown is how maximum attained energy $E_{max}$ (*left axis*) and beam energy spread (*right axis*) depend on injected charge

**Effective acceleration length**. As discussed before, electron energy is expected to scale as $n_p^{2/3}$ for a given laser power under the optimal loading condition. $E_s$ is plotted as the *dashed line* in the *inset* of Fig. 2. Related experimental data points, represented by triangles, confirm this dependency. From this we could deduce the effective acceleration length of our accelerator to be ~0.8 mm, which is shorter than the dephasing length $L_{deph} \approx 2.3$ mm predicted in the 3D nonlinear regime[22]. Hence, higher electron energies can potentially be reached by extending the accelerator to approach the dephasing length.

**Particle-in-cell simulations**. To support our results, we performed 3D Particle-In-Cell (PIC) simulations using the PIConGPU code[32, 33] (see Methods). Multiple simulations were performed taking realistic experimental parameters in order to study the beam loading effect under realistic laser-plasma dynamics, increasing only the nitrogen doping concentration in order to inject more charge into the wakefield. Although the injected charge in simulations does not reach the optimal loading condition, qualitatively the beam loading effects can be observed until this point. Figure 4a shows the temporal evolution of the injected electron energy for an example case. The driver laser beam requires the first half of the jet for focusing before ionization injection occurs. Injection is terminated due to the self-truncation effect and is followed by acceleration along ~0.9 mm

distance. This value matches our estimate from experimental data. Increasing injected charge, a clear effect on the accelerating field $E_z$ due to beam loading was observed as illustrated in Fig. 4b. For the case where a 168 pC bunch is injected, a suppression of the accelerating field $E_z$ by ~50 GV m$^{-1}$ occurs together with an easing of the accelerating field slope along the bunch compared to the weakly loaded case where a 60 pC bunch is injected. Trailing electrons being affected strongest, this field change subsequently results in a compression of the electron momentum phase-space distribution, resulting in both a reduction in $E_{max}$ and a reduction of energy spread. This effect is further illustrated in Fig. 5 where we investigate the final electron energy distribution for loads between 60 and 168 pC. In agreement with our experimental results we observe a decrease of both maximum electron energy and energy spread with increasing charge.

**Discussion**

The results presented here will have a strong impact on the parameter design of future laser-plasma accelerators. Minimization of the energy spread by means of the presented beam loading method is a crucial step toward high peak-current beams. At our laser parameters, the optimal loading condition was reached at a charge of ~300 pC within the FWHM. Although we have no direct measurement of the bunch length, using an indirect duration estimate employing a two-particle model as presented by

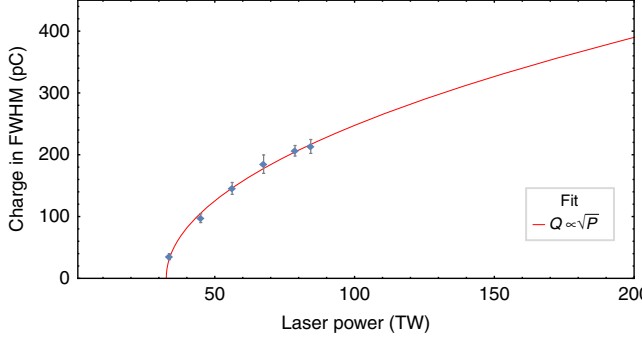

**Fig. 6** Scaling of charge within FWHM with laser power. *Circles* represent measured data points taken with a nitrogen doping of 1% at a plasma density of $3.1 \times 10^{18}$ cm$^{-3}$. *The error bars* represent the s.e.m. The *red curve* represents a fit following the expected $Q \propto \sqrt{P}$ dependency. The relative energy spread measured to be ~15% for all measurement points

Rechatin et al.[34] we could roughly determine a FWHM bunch duration of 5 fs. This is in the range of the typical measured laser wakefield bunch duration which is 4 to 6 fs[18, 19]. Thus, taking a conservative bunch length value, we estimate that our accelerator delivers a FWHM peak current of ~50 kA while operating at optimal loading condition with ~15% relative energy spread.

Working toward driving secondary light sources, future laser-plasma wakefield accelerators will combine even higher peak-current beams by operation in the nanocoulomb range as already predicted in 2002 by Pukhov and Meyer-ter Vehn[15] with boosting the energy to the tens of GeV range, supported by the tremendous advancements of laser technology pushing peak power above the petawatt level.

In Fig. 6 the correlation of the total charge to the laser peak power measured at a fixed plasma density and nitrogen doping is presented. The laser peak power was adjusted from 40 to 90 TW by an attenuator placed after the last amplifier stage. The experimental data exhibit the $Q \propto \sqrt{P}$ dependency as originally predicted by Lu et al.[22] and Gordienko and Pukhov[21]. The intersection at zero charge for a laser power of 33 TW corresponds to the injection threshold for our experimental parameters. Expanding this scaling and operating in the optimum loading condition we expect that driving a laser-plasma wakefield accelerator with a petawatt laser beam will result in high-quality electron beams with peak currents of over 150 kA. This is two to three orders of magnitude larger than can be achieved in state-of-the art large-scale accelerator facilities.

Many challenges in the field of laser-plasma acceleration still remain, besides above-mentioned scalings also transverse beam emittance. To correctly measure and further optimize beam emittance for laser wakefield acceleration in the beam loading regime is a nontrivial task and also named as one of big challenges in the field which will be the subject of further studies in this regime. Laser-plasma accelerator-driven FELs are one example of next-generation radiation sources which require low-emittance electron beams. Other next-generation radiation sources, e.g., laboratory-size beam-driven plasma accelerators, THz-radiation sources, betatron sources and inverse Compton X-ray sources, put less stringent demands on beam emittance.

The work presented here is a significant step toward the application of laser-plasma accelerators as drivers for secondary radiation sources as well as a substantial contribution on advanced accelerator concepts research. It represents an extensive systematic experimental study of beam loading by a quasi-monoenergetic bunch in the bubble regime performed with a nanocoulomb-class laser wakefield accelerator, thereby confirming the scaling theory developed by Tzoufras et al.[23]. We have

shown that in order to generate high charge beams with a small energy spread, laser-plasma accelerators have to be operated at the optimal loading condition.

## Methods

**Laser system**. The experiment was performed with the DRACO Ti:Sa chirped pulse amplification laser system operated by Helmholtz-Zentrum Dresden-Rossendorf delivering 30 fs (FWHM) pulses with an energy of 2.5 J on target. The setup allows laser diagnostics on target while the laser system is running at full power. Extra effort has been made to improve the mid field and far field of the laser beam profile as well as the correction for the angular chirp and the spectral phase. An off-axis parabolic mirror (f/20) is used to focus the laser beam onto the gas target, ~1.5 mm above the nozzle. A wavefront sensor (PHASICS SID4) in closed loop with a deformable mirror provides wavefront optimization which results to a vacuum focal spot size of 20 μm FWHM, yielding a normalized vector potential of $a_0$=2.6. Approximately 76% of the laser energy is within the beam waist ($1/e^2$ of intensity), corresponding to an estimated Strehl ratio of 0.9. This energy fraction drops to 62% at 1.5 mm away from the focus position while maintaining a Gaussian laser beam profile. Adjusting the tip tilt and gratings groove orientation of the compressor, the angular chirp of the beam was minimized to less than 0.1 μrad per nm characterized with a spectrally resolved inverted field interferometer. The spectral phase was measured with a spectral-phase interferometry for direct electric-feld reconstruction (SPIDER-A.P.E.) in parallel with a self-referenced spectral interferometry (WIZZLER-Fastlite) and optimized in a closed loop using an acousto-optic programmable dispersive filter (DAZZLER-Fastlite). Before data acquisition, the accelerator performance was further optimized by the phase correction on the second order (group velocity dispersion) and third order (third-order dispersion) at the DAZZLER. Online diagnostics situated at the experimental area were used to monitor the laser's near field and far field and its temporal stability at each shot.

**Gas target**. As plasma medium, a gas jet consisting of mixed He-N$_2$ was provided by a 3 mm supersonic de-Laval nozzle (Mach 10.4)[35] mounted on a fast gas valve (Parker 9-series). The gas-jet profile was characterized with an interferometric method[36]. It measures a density profile with a flat top region of 1.6 mm with density ramps of ~0.6 mm on both sides of the jet along the laser propagation axis. By adjusting the gas pressure at which the nozzle is operated, the gas density of the flat top profile can be varied from $10^{18}$ cm$^{-3}$ up to $5 \times 10^{18}$ cm$^{-3}$. Different nitrogen doping concentrations from 0.2% to 1.5% were achieved by using pre-mixed bottles with a doping concentration with less than 2% relative error.

**Laser-plasma accelerator**. The laser-plasma accelerator is operated in a tailored scheme of a ionization injection. STII is a combination of ionization injection with conditions such that injection only occurs over a short distance (self-truncation). Laser focus evolution and wake deformation caused by an unmatched laser spot size constrains the electron injection time leading to quasi-monoenergetic energy spectra. A more detailed description of STII can be found in ref. [28]. Operation in the ionization injection regime was confirmed, see Supplementary Note 3 and Supplementary Figs. 3 and 4.

In our experiment the STII scheme was tailored by positioning the laser vacuum focus point 1.5 mm behind the gas-jet exit. Although the vacuum focal spot of 20 μm FWHM is close to the bubble matched spot size condition (between 15 and 21 μm for experimental conditions presented in this paper), the unmatched condition required for the STII process is fulfilled by starting laser-plasma interaction in the laser intermediate field ~4.5 mm before the laser focus is reached. Focusing of the laser spot given by the geometry is enhanced by the nonlinear self-focusing effect[37] resulting in injection conditions being fulfilled at roughly the center of the gas target. Subsequent laser spot and wakefield evolution restricts the time of electron injection. The remaining acceleration length available after the point of injection is limited by the gas-jet length and thus acceleration is interrupted before the dephasing length is reached.

This tailored scheme highly requires a high laser beam quality with a large fraction of the laser energy located within the beam waist (Strehl ratio >0.9). As the interaction in this tailored scheme starts far before the focus, great care was taken to optimize the laser profile in the intermediate field around the focus together with careful spectral-phase correction (see Laser system methods above, Supplementary Note 2 and Supplementary Fig. 1).

The role of beam loading in the truncation of the injection can be excluded. For equal density and laser energy the injected charge was tuned by changing the nitrogen doping. If beam loading would be the main contributor to truncation, this would lead to continuous injection till a critical load of the bubble inhibiting further injection would be reached. Instead, we see that truncation also occurs for lower injected charges (i.e., monoenergetic feature, low dark–current background).

**Electron energy spectrometer**. After acceleration, electron beams were dispersed by a 400 mm-long permanent magnetic dipole. The field strength of the dipole was experimentally mapped using a Hall probe (Lakeshore MMTB-6J04-VG). Electron trajectories were simulated using the General Particle Tracer code employing the

measured field map. Scintillating LANEX screens (Konika Minolta OG 400) imaged onto CCD cameras (Basler acA1300-30gm) were positioned such that energy resolution is optimized with point-to-point imaging for energies up to 200 MeV[38]. Above 200 MeV the spectrometer has a readout error dominated by electron pointing. For 6 mrad pointing error this results to a readout uncertainty of $(+1.6/-1.2)$% at electron energies of 300 MeV and $(+3.1/-2.5)$% at 400 MeV. Lower pointing errors lead to a lower readout uncertainty. The spectrometer resolution and detection uncertainty is well below energy and energy spread difference found between data sets. The maximum detection energy of the spectrometer is 550 MeV. The scintillator screens were calibrated for charge at the ELBE accelerator, cross-calibrated to an Integrating Current Transformer (ICT-082-070-05:1-VAC, Bergoz Instrumentation, France) analog to the method described in ref. [39]. In order to eliminate optical imaging setup and camera efficiency effects, the cross-calibration against CLS method as described in section III B in ref. [39] was used. Cylindrical glass capsules filled with tritium and covered with scintillating material (mb-microtec trigalight T 5419-1/l green) were used as constant light sources (CLS). CLS cross-calibration between calibration setup and electron spectrometer was performed within 1 month in order to rule out tritium decay and degeneration of the scintillating material. Charge calibration was performed on scintillating screens from the same type and batch as used in the electron spectrometer.

**Particle-in-cell simulations**. PIC simulations were performed with PIConGPU[32, 33] using a 0.2.0 pre-release[40]. The simulation box used consists of $704 \times 704 \times 2{,}352$ cells with a transversal resolution of $\Delta x = \Delta y = 177$ nm and longitudinal resolution of $\Delta z = 44.3$ nm, thus resulting in a temporal resolution of $\Delta t = 0.1393$ fs. The electric and magnetic field evolution is computed via the field solver by Yee[41] while macroparticles are propagated using the particle pusher by Vay et al.[42]. The current is calculated using the Esirkepov current deposition scheme[43] with a triangular-shaped density cloud interpolation[44]. In order to incorporate ionization into the PIC cycle, similar simulations were performed using the BSI[45] and ADK[46] ionization methods. The results of the both ionization methods show good agreement for our setup, and thus BSI was selected for performance reasons. By artificially increasing the doping concentration in simulations, we were able to study beam loading effects up to 168 pC of injected charge in the peak. The plasma density was modeled according to the experimentally determined density profile of the gas target used. For the simulation presented in this paper, the electron density was set to reach $n_e = 2.62 \times 10^{18}$ cm$^{-3}$ after ionization, independent of the doping used. The laser with wavelength $\lambda = 800$ nm was modeled using a Gaussian envelope both transversally and temporally and reached a vacuum peak intensity of $a_0 = 2.8$ in focus. The pulse duration was set to $\tau = 30$ fs and the spot size to $w_0 = 19$ μm (both FWHM of intensity).

**Data availability**. The data that support the plots within this article and other findings of this study are available from the corresponding authors on reasonable request.

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

## Acknowledgements

J.P.C. and J.M.K. acknowledge LA3NET under grant agreement number 289191 from the European Unions Seventh Framework Programme. Partially funded by EUCARD2 under Grant Agreement number 312453. M.G. and A.H. acknowledge support from the European Cluster of Advanced Laser Light Sources (EUCALL) project under grant agreement No 654220 from the European Unions Horizon 2020 research and innovation programme. This project is fully supported by the Helmholtz association under program Matter and Technology, topic Accelerator Research and Development. Computations were performed on a Bull Cluster at the Center for Information Services and High Performance Computing (ZIH) at TU Dresden. We thank the ZIH for its support and generous allocations of computer time. We acknowledge M. Sobiella, C. Eisenmann and S. Grams for their technical support which enabled these experiments. We acknowledge R. Widera and all further contributors[40] to the open-source code PIConGPU for enabling our simulations.

## Author contributions

J.P.C., A.K., O.Z., J.M.K. and A.I. performed the experiment. J.P.C. and A.I. analyzed the data. R.P. and A.D. performed the simulations. M.G. and A.H. developed required PIC methods and performed tests for the simulation. M.B. supervised the simulation team. S.B., R.G., U.H., A.I., U.S. and K.Z. for laser operation and diagnostics. U.S. and A.I. provided overall supervision of the project.

## Additional information

**Competing interests:** The authors declare no competing financial interests.

