## [Peer Review File · Nature Communications]

Reviewers' comments:

Reviewer #1 (Remarks to the Author):

Laser-plasma accelerators (LPAs) are promising sources of highly relativistic electrons. Among several envisioned applications, the arguably most important ones are compact synchrotron radiation sources, in particular free-electron lasers. An FEL requires exceptionally bright electron beams. Therefore, one of the most important challenges is to show the scalability of LPA towards high spectral brightness, i.e. beams with high charge and narrow energy spread. The results presented in this manuscript are in that sense significant. They show that it is indeed possible to generate simultaneously high charge, and high beam quality (i.e. narrow energy spread). They also show that the "dark-current", or background, at low energies is small so that the majority of the charge is contained in the main peak around 250 MeV (Fig 1b). In addition, the process seems to be stable, with relatively (compared to other LPA experiments) small pulse-to-pulse fluctuations (Fig 1a, and S.1 in the supplementary). If it is proved that the measurements of the charge are correct and if the physics of the injection mechanism that was invoked to achieve this performance is properly explained, the results seem to merit publication in a high-impact journal.

1. Although it is possible that 50 kA peak current was achieved, the claim is not supported by experimental data since the pulse duration was not measured. In particular the claim "We show that the beam quality is maintained up to a peak current of 50 kA" is inaccurate since the charge, and not the peak current was measured. However, beam loading is an effect of the peak current. Could the measurements of beam loading effects be used to estimate the pulse duration and/or peak current (see e.g. Rechatin et al, NJP 12, 045023, 2010)?

2. It is claimed that localised trapping of electrons in this experiment was because of the so-called self-truncated ionisation injection (STII) mechanism. STII was described theoretically and demonstrated experimentally in previous works by other research groups. However, most experiments using ionisation injection in a uniform plasma medium have produced continuous spectra. A more detailed discussion about the physics of the injection mechanism in this experiment is therefore appropriate.

a) Why is STII expected to be dominant in this particular experiment, and what is the experimental evidence that this is indeed STII?

b) What is the injection truncation mechanism? Is it self-focusing and subsequent defocusing of an unmatched spot? Or does pulse evolution lead to an intensity significantly above the threshold for field ionisation of the inner shell electrons?

c) It is known that beam loading could also inhibit trapping of electrons in the bubble. What is the role of beam loading in the truncation of the injection?

3. Why are electrons not trapped immediately at the start of the plateau ($z \approx 1.05$ mm) in Fig 4? The two inner shell electrons are expected to field-ionise for a_0 above 1.9 and 2.8 respectively. The a_0 in the experiment was 2.6 (2.8 in the simulation), so one would expect ionisation injection to start immediately, but in Fig 4, electrons are not trapped until $z \approx 1.7$ mm, and then immediately stopped. Why?

a) Was the laser not focused at the start of plateau? If so, is the focal plane important to reproduce the conditions for this experiment?

b) Why is injection stopped so quickly? One would expect laser evolution to occur on a longer scale. For these un-matched conditions, the laser spot size and a_0 are expected to change due to periodic focusing and defocusing, and the associated deformation of the bubble might lead to injection truncation. It would be instructive to show the evolution of the laser pulse for the simulation in Fig 4 and explain why the injection starts so late and why it is truncated so quickly (or "immediately" as

written in the manuscript).

4. There are other mechanisms that could lead to peaked electron spectra, e.g. shock assisted ionisation injection (Thaury et al, Sci Rep 5, 16310, 2015). Therefore the reader might wonder if there is a shock in the supersonic gas flow. Is the interferometric method that was used for characterising the gas flow able to resolve and therefore exclude the presence of a shock? Are there other experimental evidence showing that the flow was shock free?

5. The measurement of the charge is the key result, so we have to be certain that it is correct. The measurements of the charge were performed using optical imaging of a LANEX scintillator screen onto a CCD-camera.

a) Are there any complementary measurements of the charge from the laser-plasma accelerator (e.g. using an integrating current transformer)?

b) The calibration of the scintillator was performed at the ELBE accelerator at 40 MeV. How was the scintillator response determined at 200-400 MeV?

c) The measurements by Buck et al were performed many years ago. What is the estimated effect of ageing and/or radiation damage on the scintillator response?

d) The beams produced in this experiment are very intense. Could the charge be underestimated due to saturation effects (c.f. Fig 5 in Buck et al, RSI 81, 033301, 2010)

e) The scintillator has several emission lines. Was a spectral filter used to select one of these?

f) How was the absolute response and quantum efficiency of the CCD camera determined?

g) What was the f-number and transmission of the imaging objective lens?

h) The dispersed electrons traverse the scintillator at energy-dependent angles, leading to energy-dependent effective scintillator thickness and response. Was this effect included in the analysis?

i) The scintillator emission is not isotropic. Was the camera observation angle included in the analysis?

6. Can STII be expected to inject several hundred pC over just a few hundred μm ? For the given concentrations, are there a sufficient number of nitrogen molecules available in the estimated injection region to provide that many electrons?

7. What is the resolution of the electron spectrometer, in particular in the range 200-300 MeV, where the energy spread in Fig 3 is measured?

8. The grey symbol in the inset of Fig 2 is barely visible (low contrast) when the manuscript is printed.

9. Was the LANEX screen provided by Konica Minolta?

Reviewer #2 (Remarks to the Author):

The manuscript presents experimental results on beam loading in a laser wakefield accelerator. The results are supported by 3D PIC simulations.

Beam loading occurs when a high charge electron beam, trapped in a wakefield accelerator, modifies the accelerating fields, which in turns changes the beam features. The results presented in the paper demonstrate clearly this effect. The results may be of high interest for people working on laser wakefield acceleration. Yet, several important points should be clarified before publication.

Detailed comments:

1) It is claimed in the paper that a peak current of 50 kA has been achieved, but it is not

demonstrated. The current cannot be evaluated without a measurement of the bunch duration. In the absence of such a measurement, the current may be roughly estimated, by extracting the bunch duration from simulations. From Fig. 4b, I estimate the duration to be about 13 fs. For a 300 pC charge, this leads to a current of about 20 kA. Note that in the simulation, the current is a factor of two lower, because of a lower charge. Anyway, it should not be claimed in the paper that a xx kA current has been demonstrated; the peak current can only be roughly estimated.

Lastly, it is not correct that the charge is one order of magnitude larger than the state of the art. Similar charges were obtained in [Physics of Plasmas 24, 023108 (2017)].

2) This point is connected to the previous one. The paper is widely focused on the high beam charge, which is claimed to be larger than on other LWFA facilities. My concern is that very similar experiments have been performed on several facilities, leading to lower charges. I am willing to trust the spectrometer calibration, but I would like to see in the paper an explanation for this high charge. In principle, all information required to reproduce the results should be indicated in a scientific paper. You should therefore explain how the charge can be increased by one order of magnitude on other facilities (or avoid insisting on the high charge).

3) For such a high beam charge, a transition from LWFA to beam driven acceleration could be expected. In the simulations, this does not occur because electrons are injected after a long laser propagation and the remaining plasma after the injection is rather short. However, the simulations do not guarantee that the injection occurs at this point in the experiment. Due to self-focusing I expect the injection to occur earlier at high density. In this case a transition to beam driven acceleration may be observed.

4) It is argued in the paper that the injection volume is constant. Again, this is not demonstrated. Because of the high beam charge, the bubble is strongly deformed. The higher the charge the stronger the deformation. This cavity deformation modifies the electron trajectories and the trapping conditions. The injection volume might thus depend on the charge (I would expect it to increase with the charge). Maybe simulations can show that this effect is negligible. This proof should be added to the paper.

5) Why is the energy only plotted for two different densities in the inset of Fig. 2? The scaling for the electron energy cannot be confirmed with just 2 points.

Again, why the 2 lowest densities are not plotted in Fig. 3 ?

6) While, to my knowledge, the charge is ordinarily lower in experiments than in simulations, here the opposite is observed. More generally, the agreement between the simulations and the experiment is only qualitative. This should be commented. It is also rather disappointing that simulations were run only for a density of $n=2.6 \cdot 10^{18} \text{ cm}^{-3}$, while the experiment discussion is mainly focused on largest densities ($>4 \cdot 10^{18} \text{ cm}^{-3}$) (see also comment 6).

7) As far as I understand the principle of Self Truncated Ionization Injection, the injection position and the injection length should depend strongly on the laser energy and plasma density. First, the group velocity and hence the bubble velocity depend on the plasma density (the larger the density the slower the bubble), the injection is thus easier at high density. Second, laser self-steepening, compression and self-focusing depend on both the laser intensity and the plasma density.

It is thus quite unexpected that the energy spread remains about 15% for all laser powers (as stated in the caption of Fig.6). It is also odd that the injection conditions can remain similar when the density is increased by a factor of 2 (Fig.2).

Similarly, I would also expect the acceleration length (estimated at the bottom of page 3) to depend on the density.

These issues should be discussed in the paper.

Reviewer #3 (Remarks to the Author):

Dear Editor,

High-gradient laser wakefield accelerators in a laboratory low-density plasma hold a promise to become a new compact electron injector technology that can be suitable for a wide variety of radiation sources. For a beam driven radiation source such laser-plasma accelerator must possess a combination of characteristics including energy (E), energy spread (dE/E), transverse emittance and the peak current. The paper by Couperus et al presents a study of so-called nonlinear wake or bubble regime, when a normalized laser strength, a_0 is larger than one. They show that for an ionization injection scheme an increase of the self-trapped charge results in a decrease of the energy spread due to beam loading of the wake. In a linear wake regime ($a_0 < 1$) beam loading was studied by Maja's group at LOA. The paper presents the first experimental demonstration of high-charge, low-energy spread electron beam obtained via beam loading in plasma wake and as such is of interest for the field of laser-plasma accelerators. The paper deserves publication in Nature Communications. However, major revision of the manuscript is recommended.

My specific concerns and comments are presented below.

1. The authors are ought to make statements in an abstract/paper only based on experimental measurements. I am not sure that the claimed charge of 0.5 nC is related only to the monoenergetic feature in the electron spectrum. More careful analysis of the dark current, low-energy tail and other contributions in a total measured charge is necessary because all beam current measurements (phosphor screen, ICT) represent time integrated measurements for ultrafast electron beams. Also it should be clearly stated that 50 kA peak current is based on assumptions of a 10 fs long electron bunch. The latter was never measured here.
2. The authors overlooked completely the issue of transverse emittance of the beam which is the key parameter for FEL applications. Some discussion of the beam emittance should be in the paper. Are these beams suitable for FELs that require an emittance on the order of the wavelength?
3. When I read the paper, in several places a term beam injection is applied for a self-trapped charge of plasma electrons. Typically in beam physics injection means external injection of a particle beam, in the paper there is no external injection of beam. This should be carefully introduced to avoid confusion of a reader.
4. Controllable or rather quasi-controllable self-trapping of electrons in a wake via ionization of L shell of nitrogen was first demonstrated by A. Pak et al, PRL, 104, 025003 (2010). It was simultaneously confirmed by a Michigan group. The above results should be cited.
5. Fig.2 presents a plot of energy, E versus charge but the most important dependence of beam charge versus N_2 concentration is moved to Supplemental. I recommend to combine both figures to show the key physics in the main body of paper. Please comment on an upper limit of N_2 molecules.
6. Fig. 3 can be improved by adding dE/E scaling typically used in particle beam physics. This can be done by introducing a second Y axis for dE/E .

7. It is not clear to me why the interaction length of 0.8 mm is so much smaller than the gas jet length and dephasing length. It ought to be explained.

8. How much charge is associated with the low-density ramp?

9. Section discussion almost does not contain a discussion but rather a mix of Conclusions and Discussion. Again as in 1., I recommend to give a range of peak-current values based on a more realistic estimate of the beam charge.

10. Physics of how beam loading can modify the wake and decrease the energy spread of an electron beam was discussed 30 years ago by Tom Katsouleas, (see e.g. Part Accel, 22 81 (1987)). This should be properly reflected in the paper.

Original comments as written by reviewers are printed in *italic*

To reviewer #1:

We wish to thank you for your detailed response, useful suggestions and for confirming the relevance of our work. Following your suggestion we have revised our manuscript where needed and we are confident that we can satisfactorily answer your particular questions below. You will also find a list of changes to the manuscript at the end of this document.

Laser-plasma accelerators (LPAs) are promising sources of highly relativistic electrons. Among several envisioned applications, the arguably most important ones are compact synchrotron radiation sources, in particular free-electron lasers. An FEL require exceptionally bright electron beams. Therefore, one of the most important challenges is to show the scalability of LPA towards high spectral brightness, i.e. beams with high charge and narrow energy spread. The results presented in this manuscript are in that sense significant. They show that it is indeed possible to generate simultaneously high charge, and high beam quality (i.e. narrow energy spread). They also show that the "dark-current", or background, at low energies is small so that the majority of the charge is contained in the main peak around 250 MeV (Fig 1b). In addition, the process seems to be stable, with relatively (compared to other LPA experiments) small pulse-to-pulse fluctuations (Fig 1a, and S.1 in the supplementary). If it is proved that the measurements of the charge are correct and if the physics of the injection mechanism that was invoked to achieve this performance is properly explained, the results seem to merit publication in a high-impact journal.

1. Although it is possible that 50 kA peak current was achieved, the claim is not supported by experimental data since the pulse duration was not measured. In particular the claim "We show that the beam quality is maintained up to a peak current of 50 kA" is inaccurate since the charge, and not the peak current was measured. However, beam loading is an effect of the peak current. Could the measurements of beam loading effects be used to estimate the pulse duration and/or peak current (see e.g. Rechatin et al, NJP 12, 045023, 2010)?

Indeed the 50 kA peak current reported in our manuscript is not a directly measured quantity but an estimate. Following your suggestion, we made an indirect duration estimation using the two-particle model for the bubble or blowout regime presented in Rechatin et al. (NJP 12, 045023, page 16, 2010). This model is based on the assumption that, in the unloaded case, the electron bunch is accelerated by a linearly varying wakefield. However, for the optimal loaded case, the electric field is flat between the leading and trailing electrons, thus no energy spread is added during the acceleration process. In this way the bunch length can be estimated by comparing the bunch energy spread with the electron energy in the unloaded case. This leads to equation (3) for Gaussian bunches from Rechatin et al.:

$$\frac{\Delta\gamma}{\gamma_{unld}} = 1.6 \frac{\Delta\xi}{\lambda_p}$$

With $\Delta\gamma$ the FWHM energy spread, γ_{unld} the energy reached in the unloaded case and $\Delta\xi$ the FWHM bunch duration.

Using this model we calculate our bunch duration to be around 5 fs FWHM, which gives an estimate of ~60 kA peak current in the FWHM.

Together with the typically measured laser wakefield bunch duration of 4-6 fs reported in Buck et al. (Nature Phys., 2011) and Lundh et al (Nature Phys., 2011), therefore, we are confident that 50 kA (300pC, 6 fs (FWHM)) peak current is a valid estimate.

In order to better reflect that the peak current is an estimation acquired via an indirect method, the manuscript has been adapted at several points. The abstract now states that the peak current is an

estimate (line 17). In the main body, the peak current is no longer mentioned in the results section, but is now discussed in the discussion section (lines 129-133).

2. It is claimed that localised trapping of electrons in this experiment was because of the so-called self-truncated ionisation injection (STII) mechanism. STII was described theoretically and demonstrated experimentally in previous works by other research groups. However, most experiments using ionisation injection in a uniform plasma medium have produced continuous spectra. A more detailed discussion about the physics of the injection mechanism in this experiment is therefore appropriate.

STII is a combination of ionization injection with conditions such that injection only occurs localized at a short distance (self-truncation), i.e., much less than the acceleration length. Laser beam evolution and wake deformation caused by an unmatched laser spot size constrains the electron injection time leading to quasi-monoenergetic energy spectra.

The work in our manuscript focuses on describing the beam loading effect in LWFA where the self-truncated ionisation injection mechanism was employed in order to study this effect. We are therefore convinced that a detailed discussion of the STII mechanism fits best in the methods section of our manuscript.

Accordingly, we added a new methods section (Laser plasma accelerator, lines 182-204) giving a more elaborate description of STII and the specific experimental conditions of our experiment.

a) Why is STII expected to be dominant in this particular experiment, and what is the experimental evidence that this is indeed STII?

The combination of laser and plasma density parameters chosen in our experiment is such that the wave-breaking injection threshold is not reached. This has been experimentally verified by operating under similar experimental conditions in pure helium (Supplementary figure S2 & S3). Shock assisted ionization injection is also eliminated since the gas jet was carefully characterized before the experiment and no signature of shock-front measured (see also answer to 4).

What differs us from many existing experiments is that the laser focus point was placed about 1.5 mm behind the gas jet. This allows the laser beam to interact at a relatively large spot size with the gas jet, i.e., unmatched with the bubble size as required for STII. The laser vacuum Rayleigh length is about 1.6 mm.

As it propagates inside the plasma, the laser beam will be self-focused. Once the intensity is high enough, the K-shell nitrogen electrons will be ionized and injected into the bubble. Injection is inhibited by continuation of the laser self-focusing process and bubble evolution. This is supported by our simulations (figure 4) and experimentally by the fact that our spectrum is quasi mono-energetic, which would not be the case for continuous injection.

b) What is the injection truncation mechanism? Is it self-focusing and subsequent defocusing of an unmatched spot? Or does pulse evolution lead to an intensity significantly above the threshold for field ionisation of the inner shell electrons?

The truncation mechanism is indeed given by the laser spot size evolution (see the answer to **2a**) above and the answer to question **3b**) below).

c) It is known that beam loading could also inhibit trapping of electrons in the bubble. What is the role of beam loading in the truncation of the injection?

We can exclude the role of beam loading in the truncation of the injection. For a set of plasma density and laser parameters the injected charge was tuned by changing nitrogen doping. If beam loading would be the main contributor to truncation this would lead to continuous injection until a critical load in the bubble inhibiting further injection would be reached. Instead we see that truncation also occurs

for low injected charges (i.e. mono-energetic feature, low dark-current/background), i.e., charges down to below 100 pC. We added this consideration to the manuscript at lines 200-204.

3. *Why are electrons not trapped immediately at the start of the plateau ($z \approx 1.05$ mm) in Fig 4? The two inner shell electrons are expected to field-ionise for a_0 above 1.9 and 2.8 respectively. The a_0 in the experiment was 2.6 (2.8 in the simulation), so one would expect ionisation injection to start immediately, but in Fig 4, electrons are not trapped until $z \approx 1.7$ mm, and then immediately stopped. Why?*

a) *Was the laser not focused at the start of plateau? If so, is the focal plane important to reproduce the conditions for this experiment?*

Indeed the laser vacuum focus position was not at the start of the plateau but 1.5 mm behind the gas-jet target. This focus placement is important to reproduce the conditions for this experiment. We have included this information in the new “Laser Plasma Accelerator” section under Methods (line 188).

b) *Why is injection stopped so quickly? One would expect laser evolution to occur on a longer scale. For these un-matched conditions, the laser spot size and a_0 are expected to change due to periodic focusing and defocusing, and the associated deformation of the bubble might lead to injection truncation. It would be instructive to show the evolution of the laser pulse for the simulation in Fig 4 and explain why the injection starts so late and why it is truncated so quickly (or “immediately” as written in the manuscript).*

We have followed the suggestion to update figure 4 with the laser spot size evolution (also shown below). This figure now includes the evolution of the laser field strength on-axis together with the two ionization levels of the nitrogen K-shell.

It can be seen that for our experimental parameters injection starts at the point where the first ionization level is reached and is stopped due to further evolution of the laser. Because the remaining acceleration length is short, there is no periodic laser focusing and defocusing for our experimental parameters and thus no further injection point(s) is/are reached with the exception of down-ramp injection at the end of the interaction. “Immediately” was a too strong description for the truncation effect, as it happens soon after injection but not immediate, we therefore removed this from our manuscript. The reason why injection happens so late is related to our focussing geometry and is now explained in the methods section (lines 188-195).

4. *There are other mechanisms that could lead to peaked electron spectra, e.g. shock assisted ionisation injection (Thaury et al, Sci Rep 5, 16310, 2015). Therefore the reader might wonder if there is a shock in the supersonic gas flow. Is the interferometric method that was used for characterising the gas flow able to resolve and therefore exclude the presence of a shock? Are there other experimental evidence showing that the flow was shock free?*

Our gas-jet nozzles have been designed and precision produced with spark erosion such that no shock in the supersonic gas flow is expected. This was also confirmed by the gas density measurement using a tomographic interferometry method (Couperus et al., Nucl. Instr. Meth. Phys. Res. A 830, 504-509, 2016) which is capable to resolve a shock such as presented in Thaury et al, Sci. Rep. 5, 16310, 2015. A smaller defect or damage on the nozzle could result in a shock which is small enough not to be detected by the interferometric analysis. In order to exclude this, we performed similar experiments at different nozzles produced in different batches and/or producers, resulting in the same injection behaviour.

5. *The measurement of the charge is the key result, so we have to be certain that it is correct. The measurements of the charge were performed using optical imaging of a LANEX scintillator screen onto a CCD-camera.*

a) *Are there any complementary measurements of the charge from the laser-plasma accelerator (e.g. using an integrating current transformer)?*

No complementary measurement of the charge was conducted. An integrating current transformer would not be energy selective as it would integrate all electrons originating from the accelerator over the entire energy spectrum including sub-MeV electrons.

b) *The calibration of the scintillator was performed at the ELBE accelerator at 40 MeV. How was the scintillator response determined at 200-400 MeV?*

The calibration of the scintillator was performed at the ELBE accelerator at 23 MeV. As the ELBE accelerator is not tunable toward a higher energy range, we could not experimentally determine the energy-dependent response of the scintillator. This issue is also been discussed in Buck et al., Rev. of Sci. Instr., 2010 (section II A). Simulations and measurements show that the energy deposition in scintillating screens per electron is almost constant for electron energies above 3 MeV (Y. Glinec, et al. Rev. Sci. Instrum., **77**, 103301 (2006), Hidding et al., Rev. Sci. Instrum., **78**, 083301 (2007) & Masuda et al., Rev. Sci. Instrum., **79**, 083301 (2008) Nakamura et al., Phys. Rev. ST Accel. Beams **14**, 062801, also discusses this issue in detail. In this paper a near-linear (1% less photons for every 100 MeV energy increase) response of scintillating screens was experimentally confirmed in the energy range between 106 and 1522 MeV.

c) *The measurements by Buck et al were performed many years ago. What is the estimated effect of ageing and/or radiation damage on the scintillator response?*

In order to avoid ageing and/or batch difference effects we did not use the absolute scintillator response reported in Buck et al. Instead the lanex used in experiment was calibrated relative to a constant light source analog to the method described in Buck et al. We extended *Methods, Electron energy spectrometer* section in the manuscript in order to clarify this (lines 216-222).

Ageing effects can be excluded, as the lanex calibration was performed within one month of experiment. Calibration was done with lanex screen from the same type and the same production batch as used in experiment.

Redacted

Redacted

Redacted

d) The beams produced in this experiment are very intense. Could the charge be underestimated due to saturation effects (c.f. Fig 5 in Buck et al, RSI 81, 033301, 2010)

Although the beams are very intense, the spot size on the screen of the spectrometer is such that charge densities are below 2.5 pC/mm^2 (see figure 1a). This is far below charge densities where saturation effects become relevant.

e) The scintillator has several emission lines. Was a spectral filter used to select one of these?

No spectral filter was used to select a specific emission line, both in calibration as in LWFA experiment. Instead, the spectrometer imaging system was shielded from laser, plasma emission and other background stray light.

f) How was the absolute response and quantum efficiency of the CCD camera determined?

The cross-calibration against CLS (constant light source) method as described in section III B in Buck et al. was used. This method eliminates the necessity to determine the absolute response and quantum efficiency of the CCD camera. This information has been added to the manuscript (lines 216-222)

g) What was the f-number and transmission of the imaging objective lens?

This question is closely related to the answer to the previous question **f)**. Due to the cross-calibration against CLS method, knowledge of the f-number and transmission of the imaging objective lens is not relevant for the lanex calibration. For completeness, the imaging lenses used have a f-number of 1.6.

h) The dispersed electrons traverse the scintillator at energy-dependent angles, leading to energy-dependent effective scintillator thickness and response. Was this effect included in the analysis?

Yes. A shallower angle leads to a longer effective scintillator thickness. The scintillator signal scales directly with the effective scintillator thickness. This effect was corrected for in the analysis.

i) The scintillator emission is not isotropic. Was the camera observation angle included in the analysis?

Yes. The emission of the lanex screen approximately has an Lambertian characteristic. This was taken into account depending on the camera observation angle in the analysis.

6. Can STII be expected to inject several hundred pC over just a few hundred μm ? For the given concentrations, are there a sufficient number of nitrogen molecules available in the estimated injection region to provide that many electrons?

Taking experimental conditions of a plasma density around $5 \times 10^{18} \text{ cm}^{-3}$ and a nitrogen doping concentration of 1.5%, there are 1.5×10^{17} electrons/ cm^3 available from the nitrogen inner shell (two electrons per atom, two atoms per molecule). This corresponds to $0.024 \text{ pC}/\mu\text{m}^3$. A region of $\pi \times 8^2 \times 100 \mu\text{m}^3$ (estimation derived from simulation) thus contains around 0.5 nC of charge purely from electrons originating from the nitrogen inner shell. This amount of electrons could potentially be injected in the bubble.

7. What is the resolution of the electron spectrometer, in particular in the range 200-300 MeV, where the energy spread in Fig 3 is measured?

The electron spectrometer has point-to-point imaging up to 200 MeV, giving a sub-two-percent energy resolution in that range. Above 200 MeV the spectrometer resolution depends strongly on electron pointing/divergence. Assuming 6 mrad pointing fluctuation, at 300 MeV the spectrometer has a readout uncertainty of (+1.6/-1.2)%. At 400 MeV the readout uncertainty increases to (+3.1/-2.5)%. We added this information to our manuscript (lines 210-214). The spectrometer resolution and detection uncertainty is well below energy and energy spread difference found between data sets (manuscript figure 2 & 3).

The figure below shows the readout error in more detail for different energies and pointing errors. The shaded area in this figure represents a +/- 6 mrad pointing error combined with a 200 μm source positioning error.

8. The grey symbol in the inset of Fig 2 is barely visible (low contrast) when the manuscript is printed.

The color has been changed.

9. Was the LANEX screen provided by Konica Minolta?

Yes, Konica Minolta OG 400 was used.

Original comments as written by reviewers are printed in *italic*

To reviewer #2:

We thank you for your confirmation of the relevance of our work and your helpful comments on our work presented in the manuscript. Where needed we revised our manuscript to clarify raised issues. We are confident that we can satisfactory answer your particular questions below. You will also find a list of changes to the manuscript at the end of this document.

The manuscript presents experimental results on beam loading in a laser wakefield accelerator. The results are supported by 3D PIC simulations.

Beam loading occurs when a high charge electron beam, trapped in a wakefield accelerator, modifies the accelerating fields, which in turns changes the beam features. The results presented in the paper demonstrate clearly this effect. The results may be of high interest for people working on laser wakefield acceleration. Yet, several important points should be clarified before publication.

Detailed comments:

1) It is claimed in the paper that a peak current of 50 kA has been achieved, but it is not demonstrated. The current cannot be evaluated without a measurement of the bunch duration. In the absence of such a measurement, the current may be roughly estimated, by extracting the bunch duration from simulations. From Fig. 4b, I estimate the duration to be about 13 fs. For a 300 pC charge, this leads to a current of about 20 kA. Note that in the simulation, the current is a factor of two lower, because of a lower charge. Anyway, it should not be claimed in the paper that a xx kA current has been demonstrated; the peak current can only be roughly estimated. Lastly, it is not correct that the charge is one order of magnitude larger than the state of the art. Similar charges were obtained in [Physics of Plasmas 24, 023108 (2017)].

Indeed the 50 kA peak current reported in our manuscript is not a directly measured quantity but an estimate. To further support this number we now included an indirect duration estimation using the two-particle model for the bubble or blowout regime presented in Rechatin et al. (NJP 12, 045023, page 16, 2010). This model is based on the assumption that, in the unloaded case, the electron bunch is accelerated by a linearly varying wakefield. However, for the optimal loaded case, the electric field is flat between the leading and trailing electrons, thus no energy spread is added during the acceleration process. In this way the bunch length can be estimated by comparing the bunch energy spread with the electron energy in the unloaded case. This leads to equation (3) for Gaussian bunches from Rechatin et al.:

$$\frac{\Delta\gamma}{\gamma_{unld}} = 1.6 \frac{\Delta\xi}{\lambda_p}$$

With $\Delta\gamma$ the FWHM energy spread, γ_{unld} the energy reached in the unloaded case and $\Delta\xi$ the FWHM bunch duration.

Using this model we calculate our bunch duration to be around 5 fs FWHM, which gives an estimate of ~60 kA peak current in the FWHM.

Together with the typically measured laser wakefield bunch duration of 4-6 fs reported in Buck et al. (Nature Phys., 2011) and Lundh et al (Nature Phys., 2011), therefore, we are confident that 50 kA (300pC, 6 fs (FWHM)) peak current is a valid estimate.

In order to better reflect that the peak current is an estimation acquired via an indirect method, the manuscript has been adapted at several points. The abstract now states that the peak current is an estimate (line 17). In the main body, the peak current is no longer mentioned in the results section, but is now discussed in the discussion section (lines 129-133).

Li et al., PoP 24, 023108 (2017) was not published at the moment of our submission. They report an impressive 20 kA electron beam from a laser wakefield accelerator by accelerating 620 pC of charge in a continuous spectrum up to 0.6 GeV. In the revised version of our manuscript we now reference to the article of Li et al. (lines 39-40). From 2-D simulations they found that in this case they are operating in a multi-bunch regime, with the main portion of charge located in a secondary bunch with a duration of 30 fs.

In contrast, in our work we operate a LPA in the optimal loading condition such that beam quality is optimized. In these conditions we generate around 300 pC within the FWHM of a quasi-monoenergetic (15% energy spread) beam. In these conditions the LPA can be reliably and reproducibly operated as shown by our consecutive shots.

In experiments not reported in this manuscript we have observed that if the accelerator is operated under non-ideal conditions a higher charge, and thus a probably higher current can be generated.

However, under these conditions beam quality and shot-to-shot stability is sacrificed.

The statement in our manuscript refers to operation in stable, high beam quality conditions. In comparison, Li et al. report a total charge of only 68 pC when their accelerator is operating in a quasi-monoenergetic regime.

2) This point is connected to the previous one. The paper is widely focused on the high beam charge, which is claimed to be larger than on other LWFA facilities. My concern is that very similar experiments have been performed on several facilities, leading to lower charges. I am willing to trust the spectrometer calibration, but I would like to see in the paper an explanation for this high charge. In principle, all information required to reproduce the results should be indicated in a scientific paper. You should therefore explain how the charge can be increased by one order of magnitude on other facilities (or avoid insisting on the high charge).

The main difference compared to experiments reported by groups operating at other LWFA facilities is the regime in which we operate. We operate in a tailored scheme of the self-truncated ionization injection (STII) mechanism in which the unmatched laser spot condition required for the STII mechanism is reached by starting laser-plasma interaction 4.5 mm before the laser vacuum focus position is reached. Focusing of the laser spot given by the geometry is enhanced by the non-linear self-focusing effect resulting in injection conditions being fulfilled at roughly the center of the gas target.

This tailored scheme highly requires a high laser beam quality with a large fraction of the laser energy located within the beam waist (strehl-ratio >0.9). As the interaction in this tailored scheme starts far before the focus, great care was taken for optimization of the laser profile in the intermediate field around the focus together with careful spectral-phase correction.

We added a new "Laser plasma accelerator" section to the methods section (lines 182-204) describing our scheme in more detail together with information required in order to reproduce the experiments.

Regarding the high charge, we can make a back-of-the-envelope calculation to illustrate that this scheme has the potential to inject the quantities of charge as observed in experiment: Taking experimental conditions of a plasma density around $5 \times 10^{18} \text{ cm}^{-3}$ and a nitrogen doping concentration of 1.5%, there are 1.5×10^{17} electrons/cm³ available from the nitrogen inner shell (two electrons per atom, two atoms per molecule). This corresponds to $0.024 \text{ pC}/\mu\text{m}^3$. A region of $\pi \times 8^2 \times 100 \mu\text{m}^3$ (estimation derived from simulation) thus contains around 0.5 nC of charge purely from electrons originating from the nitrogen inner shell. This amount of electrons could potentially be injected in the bubble.

3) *For such a high beam charge, a transition from LWFA to beam driven acceleration could be expected. In the simulations, this does not occur because electrons are injected after a long laser propagation and the remaining plasma after the injection is rather short. However, the simulations do not guarantee that the injection occurs at this point in the experiment. Due to self-focusing I expect the injection to occur earlier at high density. In this case a transition to beam driven acceleration may be observed.*

It is possible in LWFA that a transition to PWFA (beam driven) occurs once the electron beam starts to overtake the laser driver (approaching the dephasing limit), or once the laser driver becomes depleted. In this PWFA case, the electron beam will give its energy back to the plasma as it drives the wake, increasing the energy spread (Chou et al., Phys. Rev. Lett. 117, 144801 (2016)). In contrary we don't observe such behavior instead the energy spread keeps decreasing toward the optimum loading condition.

For the highest plasma density presented in our manuscript ($5.0 \times 10^{18} \text{ cm}^{-3}$) the dephasing length is around 2.3 mm. Due to the focusing geometry of our setup chosen for our specific study of beam loading effects (discussed in the answer to question 2)) injection does not occur until further inside the gas-jet. The gas-jet length is chosen such that the interaction is stopped before dephasing occurs. Experimentally, using E_s from equation (1), we estimate an effective acceleration length of 0.8 mm after injection for our highest plasma densities, which is below the dephasing length.

For the highest plasma density presented in our manuscript the laser depletion length is 3.2 mm, which is longer than the laser plasma interaction.

From this we can exclude a transition to PWFA in the results presented in the manuscript.

4) *It is argued in the paper that the injection volume is constant. Again, this is not demonstrated. Because of the high beam charge, the bubble is strongly deformed. The higher the charge the stronger the deformation. This cavity deformation modifies the electron trajectories and the trapping conditions. The injection volume might thus depend on the charge (I would expect it to increase with the charge). Maybe simulations can show that this effect is negligible. This proof should be added to the paper.*

The laser self-focusing and plasma dynamics are mainly influenced by the laser interaction with the background plasma. Within a dataset, plasma dynamics are kept constant and the injected charge is varied only by tuning nitrogen doping concentration (0.5%-1.5%). The injection volume is located at the front-side of the bubble, where the laser peak intensity ionizes nitrogen K-shell electrons, and then rapidly slip to the backside of the bubble, subsequently loading and/or deforming the plasma bubble. However, as remarked by the reviewer, an influence of cavity deformation on the injection length cannot be completely excluded. The figure below (left) shows that at equal plasma density and laser parameters the ionization region radius and length is independent of nitrogen doping. Although ionization occurs over a large region, injection occurs only over a limited length (see manuscript figure 4).

From our simulations we cannot extract the exact injection volume, but we can assess the beam dynamics directly after injection. The figure below (right) represents the same cases presented in figure 4 & 5 of the main manuscript and shows the electron energy distribution in the bubble evaluated right after injection ($z=1.9\text{mm}$) for different amounts of injected charge. It can be seen that injection of more charge contributes to an additional energy spread right after injection, pointing to a suspected increase of injection volume.

Comparing the energy spread right after injection and at the end of the acceleration process (figure 5 of the manuscript), we find that the contribution of energy spread from injection is small compared to the contribution of the acceleration process.

60 pC injected charge shows an absolute energy spread of ~ 10 MeV right after injection, which in this weakly loaded case grows strongly to 150 MeV during the acceleration process. In contrast, at 168 pC of injected charge the accelerating field is altered further due to beam loading, leading to only a slight

increase of the absolute energy spread during the acceleration process, from ~ 35 MeV right after injection to 60 MeV after acceleration.

Thus we can conclude that although the injection volume might vary depending on the injected charge, this effect is negligible compared to the effect from the acceleration process itself.

In the revised version of the manuscript we have added above argumentation to the supplementary and updated the text in the main body of the manuscript accordingly (lines 73-75 & line 90).

Laser field strength evolution for different nitrogen dopings from PICongPU simulations. The inner and outer region, indicated by black lines, represent the regions where the field is sufficiently high to ionize the two nitrogen K-shell electrons. Subfigure (a) corresponds to figure 4(a) in the main manuscript.

All subfigures correspond to the subfigures of figure 5 in the main manuscript, with the following charges injected in the peak due to different nitrogen dopings: (b) 103 pC, (c) 142 pC (d) 168 pC.

Electron energy histograms assessed right after injection at $z = 1.9$ mm. Subfigure (a) corresponds to figure 4(a) in the main manuscript and contains a charge of 60 pC in the main peak after acceleration.

All subfigures correspond to the subfigures of figure 5 in the main manuscript, with the following charges in the peak: (b) 103 pC, (c) 142 pC (d) 168 pC.

5) Why is the energy only plotted for two different densities in the inset of Fig. 2? The scaling for the electron energy cannot be confirmed with just 2 points.

Again, why the 2 lowest densities are not plotted in Fig.3 ?

The inset of figure 2 shows the scaling predicted by equation (1) under optimal loading conditions for our experimental conditions. In the caption this was incorrectly described as *matched conditions*, this has been corrected to *optimal loading conditions*. The two points, both representing the mean value from a set of consecutive shots, plotted in the subfigure have a charge (within the error bars) at the predicted optimum loading conditions. Other experimental data points have either lower or higher charges and thus do not represent the optimal loading condition. We agree that more than two points would have been preferable, however the inset plot is still informative to show that the scaling holds for the two datasets that are obtained at the optimal loading condition.

In the submitted version of the manuscript we only plotted the datasets in figure 3 that show the transition seen before and after the optimal loading condition. After your comment we have carefully considered our datasets and have come to the conclusion that the lowest is expected to be below to the regime border where no optimal spherical bubble is formed anymore ($a_0 < 4$). Thus, in the revised version the dataset at this density has been omitted. Figure 3 now shows all datasets also presented in figure 2.

6) While, to my knowledge, the charge is ordinarily lower in experiments than in simulations, here the opposite is observed. More generally, the agreement between the simulations and the experiment is only qualitative. This should be commented. It is also rather disappointing that simulations were run only for a density of $n=2.6 \cdot 10^{18} \text{ cm}^{-3}$, while the experiment discussion is mainly focused on largest densities ($>=4 \cdot 10^{18} \text{ cm}^{-3}$) (see also comment 6).

The full three-dimensional simulations are very demanding on resources. We use 112 K80 GPUs + 672 CPU cores on a total of 28 nodes for an average simulation time of 35 hours - this is equivalent to $\sim 24,000$ CPU hours per simulation. Therefore not all experimental parameter space could be simulated.

In our simulations we try to approach the real experimental parameters as close as possible. As we want to study the beam loading effect on the wakefield and its consequence to the final beam parameters, we avoid to vary the laser strength parameter a_0 and/or decrease the laser spot size artificially in the simulation in order to trap a higher charge, as is often done. To keep the plasma dynamics as close as possible to experiment, we artificially increased the doping concentration instead. However with the current implementation of our experimental parameters in PICongpu we were not yet able to match the charge observed in experiment. Although the injected charge in simulations does not reach the optimal loading condition, qualitatively the beam loading effects can be observed until this point.

We revised lines 110-113 such that abovementioned is better reflected in the manuscript.

As in other simulation tools, improving the PICongpu simulation code to better reflect reality is an ongoing long term-project. One of our main efforts currently being undertaken is the implementation of a more accurate representation of our experimental laser pulse using Laguerre-Gaussian modes instead of a Gaussian envelope in simulation. We mention this consideration on page 1 of the revised version of the supplementary. Iterations in this process are both resource and time demanding. Although dedicated to implement further improvements, we are of the opinion that the current state of simulations reported in our manuscript gives solid qualitative support of our experimental results.

7) As far as I understand the principle of Self Truncated Ionization Injection, the injection position and the injection length should depend strongly on the laser energy and plasma density. First, the group velocity and hence the bubble velocity depend on the plasma density (the larger the density the slower the bubble), the injection is thus easier at high density. Second, laser self-steepening, compression and self-focusing depend on both the laser intensity and the plasma density.

It is thus quite unexpected that the energy spread remains about 15% for all laser powers (as stated in the caption of Fig.6). It is also odd that the injection conditions can remain similar when the density is increased by a factor of 2 (Fig.2).

Similarly, I would also expect the acceleration length (estimated at the bottom of page 3) to depend on the density.

These issues should be discussed in the paper.

Indeed operation of a LWFA is a complicated interplay of multiple dependencies. The laser energy can have a significant influence on self-focusing, subsequent injection position and acceleration length. The purpose of figure 6 is to demonstrate experimentally that with higher laser power, more charge can be injected while not leading to a significant decrease in beam quality. The accelerator was operated close to the optimal loading condition for this broad parameter range (The optimal loaded charge depends on the laser peak power P as $Q_s \propto P^{1/2}$, see supplementary page 2). Under these experimental conditions, we found the relative energy spread remains around 15%. We interpret this phenomenon that when the laser energy is increased, a large injection volume will be generated inducing larger absolute energy spread (as we also see in experiment). However, at the same time, the wakefield strength also increases leading to higher final electron energy. Combining these two effects, a constant relative energy spread could be expected.

As described in our manuscript (line 142-144), the combination of extending this scaling together with optimization of the accelerator to optimal loading conditions opens the prospect for high-quality electrons beams at even higher peak-currents.

The injection position, injection volume and acceleration length should have a dependency on plasma density. Therefore, the beam loading effect, as presented in this manuscript, is studied at equal laser intensity and in sets of equal plasma densities.

For the estimation of the acceleration length (estimated at the bottom of page 3) the points at the optimal loading conditions were used (inset figure 2). Using equation (1) we found an acceleration length of 0.757 mm at a plasma density of $4.4 \times 10^{18} \text{ cm}^{-3}$ and 0.752 mm at $5.0 \times 10^{18} \text{ cm}^{-3}$. The left axis on the figure inset corresponds to an acceleration length of 0.755 mm.

Although the points differ (slightly) in density, we found no relevant difference in acceleration length which we suspect can be attributed to the focus geometry used.

Original comments as written by reviewers are printed in *italic*

To Reviewer #3:

We thank you for your confirmation of the relevance of our work and welcome your useful comments. Following your suggestions, we have revised our manuscript at several points and we are confident that we are able to satisfactorily answer your specific comments below. You will also find a list of changes to the manuscript at the end of this document.

High-gradient laser wakefield accelerators in a laboratory low-density plasma hold a promise to become a new compact electron injector technology that can be suitable for a wide variety of radiation sources. For a beam driven radiation source such laser-plasma accelerator must possess a combination of characteristics including energy (E), energy spread (dE/E), transverse emittance and the peak current. The paper by Couperus et al presents a study of so-called nonlinear wake or bubble regime, when a normalized laser strength, a_0 is larger than one. They show that for an ionization injection scheme an increase of the self-trapped charge results in a decrease of the energy spread due to beam loading of the wake. In a linear wake regime ($a_0 < 1$) beam loading was studied by Maka's group at LOA. The paper presents the first experimental demonstration of high-charge, low-energy spread electron beam obtained via beam loading in plasma wake and as such is of interest for the field of laser-plasma accelerators. The paper deserves publication in Nature Communications. However, major revision of the manuscript is recommended.

My specific concerns and comments are presented below.

1) *The authors are ought to make statements in an abstract/paper only based on experimental measurements. I am not sure that the claimed charge of 0.5 nC is related only to the monoenergetic feature in the electron spectrum. More careful analysis of the dark current, low-energy tail and other contributions in a total measured charge is necessary because all beam current measurements (phosphor screen, ICT) represent time integrated measurements for ultrafast electron beams. Also it should be clearly stated that 50 kA peak current is based on assumptions of a 10 fs long electron bunch. The latter was never measured here.*

In our experiment, beam charge was determined after a dispersive dipole magnet, thus enabling us to link measured charge to a specific energy. We can therefore discriminate between charge from the low energy tail and the mono-energetic peak. This is illustrated in figure 1(b), where for a single shot the charge per energy bin (pC/MeV) is plotted together with the peak FWHM region.

All experimental charges reported in our manuscript are within FWHM of the mono-energetic feature. This includes the 0.5 nC mentioned in the abstract. Due to the abstract word limit we only mention that this charge is within a mono-energetic peak. In the main text we always mention that charge is determined within FWHM. Specifically, the information that the 0.5 nC mentioned in the abstract is within the FWHM can be extracted from figure 2 and its caption.

Regarding the peak current; indeed the 50 kA peak current reported in our manuscript is not a directly measured quantity but an estimate. To further support this number we now included an indirect duration estimation using the two-particle model for the bubble or blowout regime presented in Rechatin et al. (NJP 12, 045023, page 16, 2010). This model is based on the assumption that, in the unloaded case, the electron bunch is accelerated by a linearly varying wakefield. However, for the optimal loaded case, the electric field is flat between the leading and trailing electrons, thus no energy spread is added during the acceleration process. In this way the bunch length can be estimated by comparing the bunch energy spread with the electron energy in the unloaded case. This leads to equation (3) for Gaussian bunches from Rechatin et al.:

$$\frac{\Delta\gamma}{\gamma_{\text{unld}}} = 1.6 \frac{\Delta\xi}{\lambda_p}$$

With $\Delta\gamma$ the FWHM energy spread, γ_{unld} the energy reached in the unloaded case and $\Delta\xi$ the FWHM bunch duration.

Using this model we calculate our bunch duration to be around 5 fs FWHM, which gives an estimate of ~60 kA peak current in the FWHM.

Together with the typically measured laser wakefield bunch duration of 4-6 fs reported in Buck et al. (Nature Phys., 2011) and Lundh et al (Nature Phys., 2011), therefore, we are confident that 50 kA (300pC, 6 fs (FWHM)) peak current is a valid estimate.

In order to better reflect that the peak current is an estimation acquired via an indirect method, the manuscript has been adapted at several points. The abstract now states that the peak current is an estimate (line 17). In the main body, the peak current is no longer mentioned in the results section, but is now discussed in the discussion section (lines 129-133).

2) The authors overlooked completely the issue of transverse emittance of the beam which is the key parameter for FEL applications. Some discussion of the beam emittance should be in the paper. Are these beams suitable for FELs that require an emittance on the order of the wavelength?

FEL applications have stringent beam quality requirements. Among those are peak current, energy spread and, as the reviewer remarks correctly, transverse emittance. In our manuscript we report and concentrate to a systematic study on improving peak current and energy spread as an important first-step towards FEL applications. To correctly measure and further optimize beam emittance for laser wakefield acceleration in the beam loading regime is a non-trivial task and also named as one of big challenges in the field which will be the subject of further studies in this regime.

Although an extensive discussion of the transverse beam emittance is beyond the scope of this paper, we agree with your suggestion that this is a very relevant issue. We therefore specifically mention this in the discussion of our revised manuscript as one of the challenges in the field (lines 146-149). The discussion section has also been revised such that it is properly reflected that the beam loading study presented in the manuscript is not a final solution, but a critical first step towards application for secondary light sources (lines 127-128 & 150-151).

3) When I read the paper, in several places a term beam injection is applied for a self-trapped charge of plasma electrons. Typically in beam physics injection means external injection of a particle beam, in the paper there is no external injection of beam. This should be carefully introduced to avoid confusion of a reader.

We thank you for pointing us to this unclarity. As suggested, we revised the manuscript such that it introduces that in laser-plasma acceleration, in contrast to conventional accelerators, no external injection is required and that electrons from the plasma background are injected instead (lines 32-34).

4) Controllable or rather quasi-controllable self-trapping of electrons in a wake via ionization of L shell of nitrogen was first demonstrated by A. Pak et al, PRL, 104, 025003 (2010). It was simultaneously confirmed by a Michigan group. The above results should be cited.

We thank you for mentioning this, in our original manuscript we unintentionally neglected to mention these relevant articles. In the revised version of the manuscript we now added a reference to the suggested works (A. Pak et al., PRL, 104, 025003 (2010) & C. McGuffey et al., PRL, 104, 025004 (2010)) at line 70.

5) Fig.2 presents a plot of energy, E versus charge but the most important dependence of beam charge versus N_2 concentration is moved to Supplemental. I recommend to combine both figures to show the key physics in the main body of paper. Please comment on an upper limit of N_2 molecules.

The main focus of our manuscript is the effect of beam loading in laser wakefield acceleration. This effect is shown in figure 2 and is dependent on injected charge. Although important, the dependency of injected charge on nitrogen doping shown in supplementary figure S5 is an operational parameter used to tune the accelerator such that the beam loading effect can be studied.

We feel that moving figure S5 to the main text does not add to the understanding of the beam loading effect itself and a combination of both figures would add too much complexity. Instead, we chose to explicitly point the reader to this information in the supplementary, both in the main text of the manuscript (line 89) as well as in the caption in figure 2.

Regarding an upper limit of N_2 molecules: By increasing the fraction of N_2 molecules, the amount of K-shell electrons available for injection increases. It can be expected that as the fraction of nitrogen is increased, the accelerator will operate in a different regime.

We can do a back-of-the-envelope calculation, taking experimental conditions of a plasma density around $5 \times 10^{18} \text{ cm}^{-3}$ and a nitrogen doping concentration of 1.5%, there are 1.5×10^{17} electrons/ cm^3 available from the nitrogen inner shell (two electrons per atom, two atoms per molecule). This corresponds to $0.024 \text{ pC}/\mu\text{m}^3$. A region of $\pi \times 8^2 \times 100 \mu\text{m}^3$ (estimation derived from simulation) thus contains around 0.5 nC of charge purely from electrons originating from the nitrogen inner shell. This amount of electrons could potentially be injected in the bubble.

If we would increase the fraction of nitrogen, the amount of charge injected will increase to above the optimal loading condition, thus leading to a larger energy spread.

Thus, the fraction of N_2 molecules and subsequently amount of injected charge can be increased, but this will lead to the accelerator no longer operating at the optimal loading condition. We mention this, in a less elaborate way, in the manuscript at lines 99-100.

6) Fig. 3 can be improved by adding dE/E scaling typically used in particle beam physics. This can be done by introducing a second Y axis for dE/E .

Following your suggestion we added a figure presenting dE/E to the revised manuscript.

The work presented in our manuscript studies the beam loading effect in a laser wakefield accelerator. Here a change of in energy spread is coupled with a change in mean peak energy (see supplementary figure S6). Therefore we cannot include dE/E to figure 3 by introducing a second Y-axis for dE/E . Instead a separate figure is needed as shown below.

As the absolute energy spread is decoupled from the beam loading effect on the mean peak energy, we have chosen to keep figure 3 in the main text and have added the relative energy spread to the revised version of the supplementary (figure S7).

7) *It is not clear to me why the interaction length of 0.8 mm is so much smaller than the gas jet length and dephasing length. It ought to be explained.*

We operate in a tailored scheme of the self-truncated ionization injection (STII) mechanism in which the unmatched laser spot condition required for the STII mechanism is reached by starting laser-plasma interaction 4.5 mm before the laser vacuum focus position is reached. Focusing of the laser spot given by the geometry is enhanced by the non-linear self-focusing effect resulting in injection conditions being fulfilled at roughly the center of the gas target. The remaining acceleration length available after this point is limited by the gas-jet length and thus acceleration is interrupted before the dephasing length is reached.

This tailored scheme highly requires a high laser beam quality with a large fraction of the laser energy located within the beam waist (strehl-ratio >0.9). As the interaction in this tailored scheme starts far before the focus, great care was taken for optimization of the laser profile in the intermediate field around the focus together with careful spectral-phase correction.

In the revised version of our manuscript we added a new “Laser plasma accelerator” section to the methods section (lines 182-204) describing our scheme in more detail together with information required in order to reproduce the experiments. We specifically discuss abovementioned at lines 188-195. Furthermore we have revised figure 4(a) to give a better visual representation of the injection point.

8) *How much charge is associated with the low-density ramp?*

As the downramp is located at the end of the acceleration process it contributes to the low energy background of the spectrum. For the consecutive shots shown in figure 1 and S2 (plasma density of $3.1 \times 10^{18} \text{ cm}^{-3}$) the charge associated with the low-energy background up to $\sim 100 \text{ MeV}$ is around 100 pC.

9) *Section discussion almost does not contain a discussion but rather a mix of Conclusions and Discussion. Again as in 1., I recommend to give a range of peak-current values based on a more realistic estimate of the beam charge.*

Following your remark we have thoroughly restructured the entire discussion section. Now there is a clear separation between discussion (first paragraphs) and concluding remark (last paragraph). As discussed in our answer to question **1)**, in the revised manuscript the peak current is no longer discussed in the results section, but is now discussed in the discussion section (lines 129-133) using a realistic indirect duration estimation. Furthermore we have included the issue of transverse emittance of the beam as discussed in the answer to question **2)**. (lines 146-149)

10) *Physics of how beam loading can modify the wake and decrease the energy spread of an electron beam was discussed 30 years ago by Tom Katsouleas, (see e.g. Part Accel, 22 81 (1987)). This should be properly reflected in the paper.*

We are grateful for you bringing the work of Katsouleas to our attention. In the revised manuscript we cite the linear theory for beam loading developed by Katsouleas et al. and introduce their findings that bunch shaping can be used to minimize energy (lines 47-49).

Reviewers' comments:

Reviewer #1 (Remarks to the Author):

The authors produced a detailed response to the comments by the referees, and made important changes to the manuscript.

The statement that 50 kA was achieved has been softened and is now also supported by the experimental data using the simple model.

The new laser-plasma accelerator section is useful in explaining the physics of the mechanism, and the experimental tricks that were invoked to achieve this performance. The important fact that the laser focal plane was positioned behind the gas jet was overlooked in the previous version, but is now clear. The graph showing a_0 in Fig 4 helps explaining why injection can be localised. I also appreciate the discussion on the importance of good spatial profile of the laser beam.

The description of the electron energy spectrometer is useful to acknowledge the work that was done to calibrate the electron spectrometer and the measurements of the charge.

Overall, I think the manuscript reads well, and I am willing to recommend publication.

Reviewer #2 (Remarks to the Author):

Several points have been clarified in the reply and the manuscript has been significantly improved. In the last decade, significant progresses have been made for improving the beam quality and stability in laser-plasma accelerators. However, these improvements have often been done at the price of a lower charge. Here, J.P. Couperus et al. demonstrate an increase of the charge of a stable beam by about one order of magnitude. This is an important achievement which may open the way to new applications of laser-plasma accelerators. I am therefore willing to recommend the publication of this manuscript. Yet, my opinion is tempered by a few points and I would like to see additional simulation results.

1. The current is not measured, neither supported by PIC simulations and the formula used to estimate it is based on a toy model. I therefore recommend to remove references to the current from the abstract.

2. I am not fully convinced by the estimate of the potential amount of charge that can be trapped. For this estimate, a volume of $\pi \cdot 8^2 \cdot 100 \text{ } \mu\text{m}^3$ is considered. At the considered density and for a divergence of 7 mrad, I expect a beam size of about 1-2 micrometer. I would be interested in seeing the numerical results that support the choice of this volume.

3. A beam size of several micrometers, and a divergence of 7 mrad at 350 MeV would lead to a normalized transverse emittance of a few tens of mm.mrad. This is more than one order of magnitude larger than the state of the art. This would indicate that the gain in charge has been obtained at the cost of a degraded emittance. In that case, the interest of the paper would be rather limited. I agree that an emittance measurement is difficult to achieve and probably beyond the scope of this paper. However, I cannot recommend its publication without a PIC simulation showing that low emittance beams ($< 1\text{mm.mrad}$) with peak currents in excess of 10 kA can be obtained.

Reviewer #3 (Remarks to the Author):

Dear Editor,

I have reviewed the modified version of the paper by Couperus et al. I find response of authors adequate for almost all of my comments.

However, I still consider that abstract must contain values measured in the reported experiment. Therefore, the statement of an estimated peak current of 50 kA either better be removed from the abstract or at least toned down. State-of-the art bunched beams at e.g. SLAC linear accelerator do have a >10 kA peak current of an electron beam.

To reviewer #1:

Original comments as written by reviewers are printed in *italic*.

All changes to the manuscript have been highlighted. No changes were made to the Supplementary.

The authors produced a detailed response to the comments by the referees, and made important changes to the manuscript.

The statement that 50 kA was achieved has been softened and is now also supported by the experimental data using the simple model.

The new laser-plasma accelerator section is useful in explaining the physics of the mechanism, and the experimental tricks that were invoked to achieve this performance. The important fact that the laser focal plane was positioned behind the gas jet was overlooked in the previous version, but is now clear. The graph showing a_0 in Fig 4 helps explaining why injection can be localised. I also appreciate the discussion on the importance of good spatial profile of the laser beam.

The description of the electron energy spectrometer is useful to acknowledge the work that was done to calibrate the electron spectrometer and the measurements of the charge.

Overall, I think the manuscript reads well, and I am willing to recommend publication.

Dear reviewer,

We are glad to know we could answer your questions. We appreciate the time you took for the reviewing process and your questions which led to substantial improvements of our manuscript.

To reviewer #2:

Original comments as written by reviewers are printed in *italic*.

All changes to the manuscript have been highlighted. No changes were made to the Supplementary.

Several points have been clarified in the reply and the manuscript has been significantly improved. In the last decade, significant progresses have been made for improving the beam quality and stability in laser-plasma accelerators. However, these improvements have often been done at the price of a lower charge. Here, J.P. Couperus et al. demonstrate an increase of the charge of a stable beam by about one order of magnitude. This is an important achievement which may open the way to new applications of laser-plasma accelerators. I am therefore willing to recommend the publication of this manuscript. Yet, my opinion is tempered by a few points and I would like to see additional simulation results.

Dear reviewer,

We thank you for your contribution to the review process and the improvements we could implement to our manuscript following your comments. We are glad that we were able to answer your questions from the previous round. Please find a point-to-point response to newly raised issues below.

1. The current is not measured, neither supported by PIC simulations and the formula used to estimate it is based on a toy model. I therefore recommend to remove references to the current from the abstract.

Following your recommendation, we limited the discussion of the peak current to the discussion section of our manuscript and removed the reference to the peak current estimate from the abstract.

2. I am not fully convinced by the estimate of the potential amount of charge that can be trapped. For this estimate, a volume of $\pi \cdot 8^2 \cdot 100 \text{ } \mu\text{m}^3$ is considered. At the considered density and for a divergence of 7 mrad, I expect a beam size of about 1-2 micrometer. I would be interested in seeing the numerical results that support the choice of this volume.

The injection radius differs from the accelerated bunch radius and is considerably larger. The injection volume where K-shell electrons are ionized is located at the front of the bubble around the laser peak intensity. After trapping and during acceleration, the strong transverse fields of the bubble confine these electrons to a matched beam size considerably smaller than the injection volume.

The choice for the 8 μm radius of the injection volume used for the estimate of the potential amount of charge is based on numerical PIC simulations, whose results are shown in supplementary figure S8 (also shown below). In the answer to question 3 you will find PIC results showing the accelerated bunch size, which was found to be around 1 μm .

FIG. S.8: Laser field strength evolution for different nitrogen doping concentrations from PIConGPU simulations. The inner and outer region, indicated by black lines, represent the regions where the field is sufficiently high to ionize the two nitrogen K-shell electrons. Subfigure (a) corresponds to figure 4(a) in the main manuscript. All subfigures correspond to the subfigures of figure 5 in the main manuscript, with the following charges injected in the peak due to different nitrogen dopings: (b) 103 pC, (c) 142 pC (d) 168 pC.

3. A beam size of several micrometers, and a divergence of 7 mrad at 350 MeV would lead to a normalized transverse emittance of a few tens of mm.mrad. This is more than one order of magnitude larger than the state of the art. This would indicate that the gain in charge has been obtained at the cost of a degraded emittance. In that case, the interest of the paper would be rather limited. I agree that an emittance measurement is difficult to achieved and probably beyond the scope of this paper. However, I cannot recommend its publication without a PIC simulation showing that low emittance beams (< 1mm.mrad) with peak currents in excess of 10 kA can be obtained.

Indeed, the beam emittance is an important parameter for specific next-generation compact light-sources driven by LWFA generated electron beams. As discussed in the previous round with reviewer 3, this is specifically the case for FELs. As stated, accurate experimental determination of beam emittance and in particular slice emittance is a non-trivial task and is named as one of the big challenges in the field which is beyond the scope of the beam-loading study presented in our manuscript.

The back-of-the-envelope calculation we assume you used for estimating the normalized transverse emittance uses experimentally measured parameters which are extracted from the original manuscript. To our opinion this method has significant conceptual shortcomings as we will discuss below. Furthermore, we have to assume that due to a misunderstanding in the dataset used (as also discussed below) you deduced a value of tens of mm.mrad, significantly overestimating the value of 3.4 to 6.8 mm.mrad we achieve by applying this back-of-the-envelope approach with a consistent set of experimental data.

Even if the emittance is a few mm.mrad, we respectfully disagree with your statement that such a transverse emittance significantly limits the general interest in the paper. FELs are one example of next generation radiation sources which require low (slice) emittance. However, there are numerous other applications (e.g. laboratory-size beam-driven plasma accelerators, intense single cycle THz-radiation sources, betatron sources and inverse Compton x-ray sources) that place less stringent demands on beam emittance and the research presented in our manuscript is a vital step towards increasing the brightness of these sources for wide ranging applications.

Following the suggestion from reviewer 3 in the previous review-round, we do discuss the transverse emittance in the discussion section of our manuscript. Even though we are not able to report on beam emittance measurements directly, we find that this paragraph adds to the reader's understanding and creates awareness that for certain applications further research is still required. This does not diminish

the value of the core message of our work on the beam loading effect. We have expanded this paragraph in the manuscript for the reader's understanding, including above argument (lines 150-153) Our manuscript addresses one of the current major topics in the plasma accelerator community, namely increasing the charge and peak current output of plasma accelerators. The work reported is crucial for both application at secondary radiation sources as well as for future research as it demonstrates that LWFAs operating at high charge and peak current must be operated at the optimal loading condition. This includes the next research step aimed on measurement and minimization of the transverse emittance for FEL applications.

Nonetheless we would like to discuss the normalized beam emittance in more detail. The back-of-the-envelope model mentioned before utilizes source size, electron energy and extracted beam divergence ($\epsilon = \gamma \cdot \sigma_r \cdot \sigma_{div}$). Even though the injection volume is larger, the 1-2 micrometer beam size estimate you made in question 2 is realistic and fits with values we found in our simulations (see below). The 7 mrad divergence was reported for an electron energy of 250 MeV, not 350 MeV (see paper). Thus employing this simple method we find a transverse emittance of 3.4 to 6.8 mm.mrad instead of a few tens of mm.mrad.

However, this is a strongly simplified model, ignoring correlation terms¹ and emittance growth due to free-space propagation after acceleration. A currently discussed problem within the LWFA community is that the strong divergence combined with the relatively large energy spread leads to a strong transverse emittance growth once laser accelerated beams exit the plasma^{2,3}. In order to limit emittance growth it is required that strong focusing elements matched to the beam parameters are positioned close the accelerator exit^{4,5}. In our case the large bunch charge combined with the energy spread causes that the divergence (i.e. the geometric spot size at a distance) measured after a long free-space propagation does not directly reflect the angular spread at the plasma exit. As a consequence the model mentioned above strongly overestimates the transverse emittance¹ making the 3.4 to 6.8 mm.mrad an inaccurate, but upper limit, estimate.

Expansion of the manuscript to include an accurate discussion of transverse emittance does not fit the core message and would be a study on its own. As mentioned above, accurate measurement and optimization of beam emittance is a non-trivial task which is beyond the scope of the beam-loading study presented in our manuscript. We seem to be in agreement on this.

¹ Curcio, A., Anania, M., Bisesto, F., Chiadroni, E., Cianchi, A., Ferrario, M., ... Zigler, A. (2017). Trace-space reconstruction of low-emittance electron beams through betatron radiation in laser-plasma accelerators. *Physical Review Accelerators and Beams*, 20(1), 12801. <http://doi.org/10.1103/PhysRevAccelBeams.20.012801>

² Antici, P., Bacci, A., Benedetti, C., Chiadroni, E., Ferrario, M., Rossi, A. R., ... Bacci, A. (2012). Laser-driven electron beamlines generated by coupling laser-plasma sources with conventional transport systems. *Journal of Applied Physics*, 112(44902). <http://doi.org/10.1063/1.4740456>

³ Migliorati, M., Bacci, A., Benedetti, C., Chiadroni, E., Ferrario, M., Mostacci, A., ... Antici, P. (2013). Intrinsic normalized emittance growth in laser-driven electron accelerators. *Physical Review Special Topics - Accelerators and Beams*, 16(1), 11302. <http://doi.org/10.1103/PhysRevSTAB.16.011302>

⁴ Barber, S. K., Schroeder, C. B., Van Tilborg, J., & Leemans, W. P. (2017). Transport and phase-space manipulation of laser-plasma accelerated electron beams using active plasma lenses Experimental characterization of active plasma lensing for electron beams Transport and Phase-Space Manipulation of Laser-Plasma Accelerated Electron Beams Using Active Plasma Lenses. *AIP Conference Proceedings Applied Physics Letters*, 40006(101), 20002–40008. <http://doi.org/10.1063/1.4975840>

⁵ Couprie, M. E., Labat, M., Evain, C., Marteau, F., Briquez, F., Khojayan, M., ... Loulergue, A. (2016). An application of laser-plasma acceleration: towards a free-electron laser amplification. *Plasma Physics and Controlled Fusion*, 58(3), 34020. <http://doi.org/10.1088/0741-3335/58/3/034020>

Our manuscript reporting on an experimental study on beam loading at high bunch charge, we consider it inappropriate to extend the discussion on emittance under conditions based on novel and demanding simulations alone and without the proper experimental support.

This said, we wish to share values we extracted from our existing PIC simulations for the sake of discussion. However, we find your specific demand for a value of < 1 mm.mrad rather ambitious for several reasons.

Firstly, from an experimental point of view the value you ask for is rather low. Although a few experimental reports exist where a < 1 mm.mrad normalized transverse emittance is reported for plasma accelerators^{1,6}, these are reported at accelerators with significantly less charge. For the high charge as in our accelerator together with the fact that we did not specifically optimized the emittance, we find an estimate of a few mm.mrad on par and within the same order of magnitude of the current state of the art conventional RF and laser plasma accelerators. For example, Li et al.⁷, report a normalized transverse emittance at energy slices between 6 and 31 mm.mrad.

Secondly, in the current state of the art, particle-in-cell simulations of LWFA rely on a variety of numerical steps and underlying approximative models. Until very recently, for example, most PIC codes grossly overestimated the transverse emittance and bunch diameter due to numerical influences from a combination of numerical dispersion and numerical Cherenkov radiation. Although work on Maxwell-solvers to mitigate this problem has been published recently (Jalas et al.⁸, Lehe et al.⁹), these schemes either do not entirely eliminate the numerical problem or influence other properties of the final electron beam such as the final energy and energy spread as they alter the temporal evolution of the accelerating gradients. This is an ongoing field of research and as such we strongly believe that there is not a single PIC code currently available that has fully predictive power with regard to all relevant electron beam parameters simultaneously. This means that with a specific choice of models and numerical schemes we can optimize on predicting, e.g. the transverse emittance, but then fail in predicting other beam parameters with the same accuracy. This makes a completely accurate matching of simulation to experiment a task that cannot be justified with regard to the ongoing research on driving the capabilities of PIC codes towards full quantitative prediction of all beam parameters of LWFA-driven beams. Nevertheless, we want to address the understandable concern of the referee that transverse emittance is a beam property that will be important for future applications of such beams. With this in mind we are thus willing to share the information on emittance values given by the PIC simulations. As the Maxwell solvers used in this simulation do not fully suppress numerical Cherenkov radiation, these values can only be regarded as upper bounds to the emittance expected in experiment.

In the figures below you will find several plots from PIC simulations. These results are based on the same simulation as presented in main manuscript. Specifically, these plots represent the same simulation as shown in figure 5d representing the 168 pC case. Plots below represent data evaluated at $z=2.7$ mm, that is at the end of the acceleration process and already in the gas density downramp.

⁶ Golovin, G., Banerjee, S., Liu, C., Chen, S., Zhang, J., Zhao, B., ... Umstadter, D. (2016). Intrinsic beam emittance of laser-accelerated electrons measured by x-ray spectroscopic imaging. *Scientific Reports*, 6(April), 24622. <http://doi.org/10.1038/srep24622>

⁷ Li, Y. F., Li, D. Z., Huang, K., Tao, M. Z., Li, M. H., Zhao, J. R., ... Chen, L. M. (2017). Generation of 20 kA electron beam from a laser wakefield accelerator. *Physics of Plasmas*, 24(2), 23108. <http://doi.org/10.1063/1.4975613>

⁸ Jalas, S. et al. Accurate modeling of plasma acceleration with arbitrary order pseudo-spectral particle-in-cell methods. *Phys. Plasmas* 24, 1–7 (2017).

⁹ Lehe, R., Lifschitz, A., Thaury, C., Malka, V., & Davoine, X. (2013). Numerical growth of emittance in simulations of laser-wakefield acceleration. *Physical Review Special Topics - Accelerators and Beams*, 16(2), 1–8. <http://doi.org/10.1103/PhysRevSTAB.16.021301>

Figure A1 PIC simulation result showing the transverse electron distribution evaluated at $z=2.7$ mm. The white line represents the 1σ contour.

Figure A2 PIC simulation result showing the electron phase space distribution evaluated at $z=2.7$ mm. The white line represents the 1σ contour.

Figure A3 PIC simulation result showing the longitudinal electron distribution evaluated at $z=2.7$ mm. The orange area represents electrons within 2σ transversally and within the $\Delta\tau=15.2$ fs window and contains 158.3 pC of charge.

As shown in figure A1, we find a transverse bunch area of $2.81 \mu\text{m}^2$ (rms), corresponding to a radius of $0.95 \mu\text{m}$. The simulation gives a transverse emittance of 4.42 mm.mrad as shown in figure A2. At this point we want to stress again that we consider this an overestimate. In this specific case the charge was found to be 158.3 pC within 15.2 fs , as shown in figure A3, resulting in a 10.4 kA current. Please note that in experiment bunches with a larger charge were observed.

Using the back-of-the envelope calculation, we show that the normalized beam emittance is not as large as originally feared. From our experimental findings alone, it is not possible to determine an exact value of the emittance, neither is it possible to exclude emittances below 1 mm.mrad . Nevertheless, finding an upper limit of 3.4 to 6.8 mm.mrad , our accelerator would be a suitable driver for many secondary radiation sources.

Furthermore, we want to emphasize that our study reports on the demonstration of the beam loading effect and how longitudinal beam parameters are influenced. This study is a crucial step for both application at secondary radiation sources as well as for future research as it demonstrates that LWFA's operating at high charge and peak currents must be operated at the optimal loading condition in order to retain high beam quality.

To reviewer #3:

Original comments as written by reviewers are printed in *italic*.

All changes to the manuscript have been highlighted. No changes were made to the Supplementary.

Dear editor,

I have reviewed the modified version of the paper by Couperus et al. I find response of authors adequate for almost all of my comments.

However, I still consider that abstract must contain values measured in the reported experiment. Therefore, the statement of an estimated peak current of 50 kA either better be removed from the abstract or at least toned down. State-of-the art bunched beams at e.g. SLAC linear accelerator do have a >10 kA peak current of an electron beam.

Dear reviewer,

We thank you for your contribution to the review process and the improvements we were able to implement to our manuscript following your comments. Following your latest remark we agree that the abstract should focus on the main achievements of the experiment only. We now only treat the peak current in the final discussion section and removed the reference to an estimated peak current, including the comparison with conventional accelerators, from the abstract.

REVIEWERS' COMMENTS:

Reviewer #2 (Remarks to the Author):

I have been convinced by the authors response and I recommend publication.